# SurgRe4DGS: Open-Vocabulary Vision-Language Gaussian Splatting for Retrievable 4D Surgical Scene Reconstruction

## Abstract

Reconstructing dynamic 4D models from surgical workflows is crucial for robotic-assisted surgery, real-time navigation, and postoperative analysis. However, real-world challenges encountered during surgery, such as tissue deformations and occlusions, significantly increase the difficulty of reconstruction and semantic perception. In this work, we introduce *SurgRe4DGS*, a novel framework that jointly realizes 4D surgical scene reconstruction and open-vocabulary instrument retrieval through synergistic vision–language Gaussian representations. Particularly, we leverage visual clues to enhance color representations through spatial-color blending with temporal modulation, while language features are instilled with hybrid injection strategies via semantic map rendering and weighted top-K aggregation with contrastive learning. With a dual-branch mechanism for vision-language context integration, *SurgRe4DGS* ensures precise cross-modal alignment and semantic consistency in deformable scenarios. To fill the gap in existing benchmarks for language-driven multimodal 4D reconstruction and open-vocabulary instrument retrieval, we introduce *StereoLung15K*, a pioneering surgical dataset with 26 binocular thoracoscopic sequences, over 15,000 annotated frames, and multimodal (RGB/depth/text) support at 30 FPS, enabling joint reconstruction and text-guided retrieval tasks. Extensive experiments on two public datasets, StereoMIS and EndoNerf, as well as our *StereoLung15K*, demonstrate that *SurgRe4DGS* achieves state-of-the-art performance in terms of reconstruction fidelity and retrieval precision. Our code and data will be publicly available.

## 1 Introduction

High-quality 3D surgical scene reconstruction provides surgeons with intraoperative augmented reality, thereby reducing surgical risks and advancing clinical applications (Chen et al., 2025a). In dynamic surgical procedures, 4D scene reconstruction proves to be more practical for capturing tissue deformation, instrument motion, and interactions over temporal changes, facilitating continuous spatial modeling within video streams through precise spatiotemporal representations (Liu et al., 2025). However, real-world surgical environments pose significant challenges, including rapid and irregular deformations, frequent occlusions, and unexpected environmental factors like smoke, lighting, and blood, which further complicate 4D scene modeling and semantic understanding. Despite progress in recon-

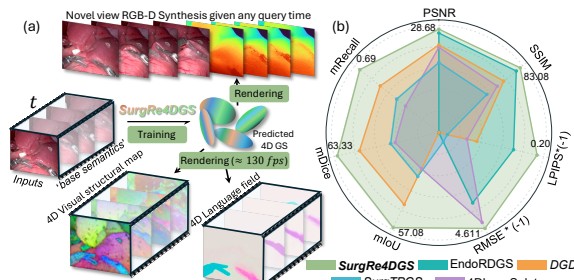

Figure 1: Workflow illustration (a), and average performance comparison on three datasets (b).

struction techniques (Gao et al., 2025), existing methods primarily focus on geometric fidelity and novel view synthesis, often overlooking the semantic perception of scene entities (*e.g.*, surgical instruments), which is essential for understanding intraoperative interactions and states in clinical operations. Fundamentally, scene reconstruction and semantic understanding are intrinsically associated (Liao et al., 2025). On the one hand, semantic context, especially from vision–language models that support open-vocabulary querying, can guide reconstruction by highlighting critical regions and anatomical boundaries in dynamic scenes. On the other hand, scene reconstruction, particularly precise 4D reconstruction, provides rich spatiotemporal priors that enhance instrument retrieval and perception in complex environments. Together, these reciprocal benefits facilitate a synergistic alignment between geometric and semantic representations, making 4D reconstruction a well-suited foundation for semantic scene understanding in surgical settings.

Recent 4D reconstruction works (Yang et al., 2024b;a; Shan et al., 2025; Liu et al., 2025; Gao et al., 2025) are dedicated to developing flexible deformation models for time-dependent scene dynamics. These approaches, however, prioritize novel view synthesis and geometric reconstruction, often neglecting semantic representation, which limits their performance potential in real-world applications. More recently, Huang et al. (2025) attempts to incorporate text-promptable GS for dynamic semantic understanding, but it restricts queries to predefined forms and lacks open-vocabulary retrieval for novel queries. Other language-embedded methods (Labe et al., 2024; Li et al., 2025b; Fiebelman et al., 2025) for 4D scenes excel in object-centered settings yet struggle in surgical environments due to severe occlusions and extensive deformations. Practically, human anatomy variability complicates Gaussian semantic optimization, as temporal drift in Gaussian-attached semantics leads to inconsistent representations across frames. This disrupts spatial coherence between visual cues and semantic attributes, causing misalignments in vision-language embedding spaces.

To address above challenges, we propose *SurgRe4DGS*, an innovative framework that leverages unified visual-lingual Gaussian representations for synergistic optimization of 4D surgical scene reconstruction and instrument retrieval (Fig. 1(a)). By integrating visual and linguistic cues in a unified representation space, it not only delivers precise instrument localization but also ensures that reconstructions accurately align with instrument positions and spatial relationships. Specifically, we augment each Gaussian primitive with partitioned feature vectors represented in visual and linguistic subspaces, with a dual-branch mechanism optimizing these subspaces synergistically. The visual branch distills spatial structure priors from the 2D large vision model (*e.g.*, SAM (Kirillov et al., 2023)) to enhance anatomical boundary sensitivity through feature blending and temporal modulation. The language branch injects textual semantics via semantic map rendering for coarse alignment and top-K aggregation with contrastive learning for fine-grained refinement. These integrated representations are rendered into holistic feature maps, facilitating seamless cross-modal refinement without explicit supervision, thus ensuring alignment between vision and language contexts and semantic consistency under dynamic conditions. Moreover, to support language-driven instrument-retrievable 4D reconstruction, we build *StereoLung15K*, a large-scale real surgical scene dataset with 26 binocular thoracoscopic sequences and over 15,000 annotated frames, bridging gaps in high-frame-rate instrument-text-4D retrieval benchmarks. Extensive experiments validate the superiority of *SurgRe4DGS* in 4D reconstruction and free-form instrument retrieval tasks (Fig. 1(b)). In summary, the main contributions of this work are:

- We propose *SurgRe4DGS*, a unified framework that jointly optimizes 4D surgical scene reconstruction and open-vocabulary instrument retrieval via synergistic vision-language Gaussian representations, enabling precise localization and spatially coherent reconstructions for enhanced intraoperative understanding.
- We develop a dual-branch mechanism that augments Gaussian primitives with a unified latent space, where the visual branch enhances color fidelity through spatial-color blending with temporal modulation, while the language branch employs hybrid injection via semantic map rendering and top-K contrastive alignment for robust vision-language consistency across views and time.
- We introduce *StereoLung15K*, a pioneering multi-modal surgical reconstruction dataset featuring 26 binocular thoracoscopic sequences and over 15,000 annotated frames with RGB/depth/text modalities at 30 FPS.
- Extensive experiments on two public datasets and *StereoLung15K* demonstrate that our approach significantly outperforms existing methods in 4D reconstruction and open-vocabulary instrument retrieval tasks.

Table 1: Comparison between *StereoLung15K* and existing real-world surgical dynamic reconstruction and instrument-oriented datasets. "≈" stands for the available frames for reconstruction.

| Dataset | Statistics | | | Attributes | | | | Tasks | |
| --- | --- | --- | --- | --- | --- | --- | --- | --- | --- |
| | No. Seq. | No. frames | No. Tool cls | Tool Ins. Anno. | Tool Text. Anno. | Anno. at FPS | Scene | 4D Recons. | Text-driven 4D Tool Retri. |
| EndoVis17 (Allan et al., 2019) | 10 | 3,000 | 7 | ✔ | ✗ | 1 | Dynamic | ✗ | ✗ |
| EndoVis18 (Allan et al., 2020) | 19 | 5,700 | 5 | ✔ | ✗ | 1 | Dynamic | ✗ | ✗ |
| EndoNeRF (Wang et al., 2022) | 5 | 807 | 2 | ✗ | ✗ | 15 | Dynamic | ✔ | ✗ |
| StereoMIS (Hayoz et al., 2023) | 11 | ≈3,000 | 3 | ✗ | ✗ | 30 | Dynamic | ✔ | ✗ |
| SCARED (Allan et al., 2021) | 9 | ≈1,900 | 0 | ✗ | ✗ | 30 | Static | ✔ | ✗ |
| Hamlyn (Yang et al., 2024a) | 7 | 2,107 | 2 | ✗ | ✗ | 30 | Dynamic | ✔ | ✗ |
| *StereoLung15K* | **26** | **15,787** | **7** | ✔ | ✔ | 30 | Dynamic | ✔ | ✔ |

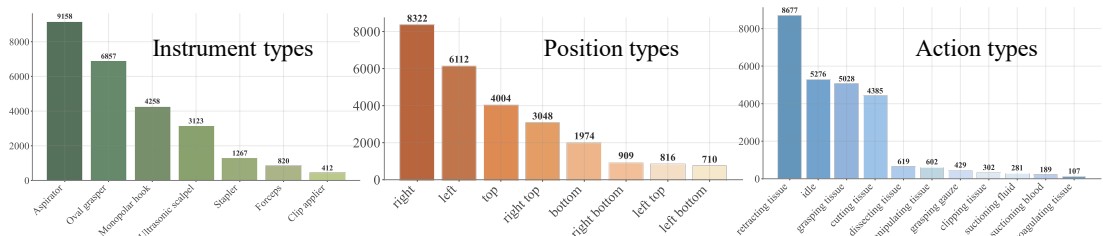

Figure 2: Statistics of text semantics components in *StereoLung15K*.

## 2 STEREOLUNG15K DATASET

Current methods for surgical scene reconstruction are limited by the scarcity of high-frame-rate data for 4D modeling and open-vocabulary querying, resulting in temporal blurring and poor semantic alignment. To this end, we introduce *StereoLung15K*, comprising frames at 30 FPS with rich tool labels with textual descriptions and multimodal annotations for 4D retrievable surgical scene reconstruction.

**Data Collection and Annotation.** *StereoLung15K* is built on real binocular thoracoscopic videos of pulmonary surgeries, including lobectomy and segmentectomy. We collected 26 video sequences from partner hospitals to capture surgical complexities, *e.g.*, multi-type instruments, bleeding, smoke, occlusions, and tissue deformations. Each sequence offers synchronized binocular views with calibrated intrinsics for stereo depth estimation, captured at 30 FPS to minimize temporal artifacts. Annotations are performed for each frame at 30 FPS by four surgeons, with two for primary labeling and two for verification, covering instrument instance-level masks and textual semantics. For instrument mask, we employ a semi-automated workflow integrating AI-assisted tools with expert manual labeling (Chen et al., 2025b). For text semantics, we adopted a structured strategy with three key attributes per instrument (<"*type*","*action*","*position*">), *e.g.*, <*"Grasper", "cutting tissue", "top"*>. These can be algorithmically combined into natural descriptions, enabling scalable open-vocabulary querying while ensuring clinical relevance.

**Dataset Statistics and Analysis.** *StereoLung15K* includes 26 sequences with 15,787 annotated frames at 1920×1080 resolution, spanning 7 instrument categories, 11 actions, and 8 position attributes (Fig. 2), for a total of 25,895 instances (average 607.2 frames per sequence, 1.64 instances per frame). Tab. 1 compares *StereoLung15K* with existing surgical scene reconstruction and instrument-oriented benchmarks. Datasets like EndoVis17/18 (Allan et al., 2019; 2020) offer instrument annotations, but at low frame rates (1 FPS) and without textual semantics, which limits dynamic language-driven tasks. Others, such as EndoNeRF (Wang et al., 2022), StereoMIS (Hayoz et al., 2023), SCARED (Allan et al., 2021), and Hamlyn (Yang et al., 2024a), focus on reconstruction yet lack semantic attributes like instrument instance mask or textual annotations. In contrast, *StereoLung15K* excels in scale (most sequences and labeled frames), comprehensive multimodal

Figure 3: Overview of proposed *SurgRe4DGS* for 4D surgical scene reconstruction and instrument retrieval. The framework is trained by optimizing a unified dual-branch latent vector ($z_i = [f_i^v, f_i^l]$) for each 3D Gaussian primitive. The visual branch ($f_i^v$) is trained to match spatial-structural features from a 2D vision foundation model, while the language branch ($f_i^l$) is trained on MLLM-enriched semantic embeddings via our novel hybrid semantics injection scheme. Both branches utilize temporal modulation to ensure spatio-temporal consistency. During inference, the model renders high-fidelity novel views or, given a natural language query, renders the learned language map ($\hat{\mathcal{F}}^l$) to get the activation map for query localization.

annotations at 30 FPS, and support for 4D reconstruction and language-driven instrument retrieval. More sample exhibitions and information statistics can be found in Appendix. C.

## 3 PROPOSED METHOD

### 3.1 PRELIMINARIES FOR DYNAMIC GAUSSIAN SPLATTING

Vanilla 3DGS (Kerbl et al., 2023) represents a scene as a set of 3D Gaussians $\mathcal{G} = \{g^i\}_{i=1}^{\mathcal{N}}$, each with position $\mu^i \in \mathbb{R}^3$, covariance matrix $\sum^i \in \mathbb{R}^{3 \times 3}$, opacity $o^i \in \mathbb{R}$, and color features $c^i \in \mathbb{R}^{d_c}$. Dynamic Gaussian extends this to 4D scenes by conditioning parameters on time $t$ via a deformation field $D(t)$. The color $\hat{C}$ and depth $\hat{D}$ at pixel $p$ are computed by $\alpha$-blending $n$ sorted deformed Gaussians that overlap $p$:

$$\hat{C}(p) = \sum_{i=1}^{n} c_i o_i G_i \prod_{j=1}^{i-1}(1 - o_j G_j), \quad \hat{D}(p) = \sum_{i=1}^{n} d_i o_i G_i \prod_{j=1}^{i-1}(1 - o_j G_j), \tag{1}$$

where $d_i$ is the $z$-depth in view space, and $G_i = e^{-(p-\mu_{2D}^i)(\sum_{2D}^i)^{-1}(p-\mu_{2D}^i)/2}$. The 2D projections $\mu_{2D}^i$ and $\sum_{2D}^i$ are analytically derived from camera extrinsics $T_t$, intrinsic $K$ and deformation parameters $D(t)$.

## 3.2 SURGRE4DGS FRAMEWORK

Fig. 3 illustrates the overview of our framework. Given a surgical video $\mathcal{I} = \{I_t\}_{t=1}^{T}$, we aim to train a model $\Phi$ that reconstructs the underlying deformable 4D scene with instrument-retrievable features, represented as vision-language enhanced dynamic 3D Gaussians. During training, we first extract pixel-aligned instance-level 4D semantic features using a multimodal large language model (MLLM) (Sec. 3.2.1). Then, we model the dynamic Gaussians via integrated spatial-appearance features and a hybrid language injection strategy, which incorporates spatial-temporal consistency constraints and a semantic contrastive scheme (Sec. 3.2.2). This setup enables open-vocabulary instrument queries during inference (see Appendix. D).

### 3.2.1 INSTRUMENT-WISE SEMANTIC ENRICHMENT VIA MLLM

Robust open-vocabulary instrument retrieval and semantic-guided 4D reconstruction require enriching basic annotations with detailed context-aware descriptions. By leveraging MLLM (Bai et al., 2025) to generate refined descriptions for enhancing unified visual-language Gaussian representations, it can support free-form querying and provide priors to regularize dynamic Gaussians. Unlike directly generating text from object-wise video to learn 4D language fields (Li et al., 2025b), we ground the enrichment in instrument masks and structured semantics, enabling features better aligned with visual dynamics to handle extreme conditions.

During training, for each frame $I_t$ in given frames $\mathcal{I}_{tr} = \{I_1, I_2, \ldots I_T\} \in \mathbb{R}^{3 \times H \times W}$, we utilize annotated instrument masks $\mathcal{M}_{o_i t} \in M_{Ot} = \{\mathcal{M}_{o_1 t}, \mathcal{M}_{o_2 t}, \ldots, \mathcal{M}_{o_m t}\}$, and the corresponding basic semantics $\mathcal{S}_{o_i t} \in S_{Ot} = \{\mathcal{S}_{o_1 t}, \mathcal{S}_{o_2 t}, \ldots, \mathcal{S}_{o_m t}\}$ to construct enriched descriptions, where $m$ denotes the number of instruments. We compose initial expressions $e_{o_i t}$ from basic semantics (*e.g.*, *"Oval grasper is idle on the right"*) and feed them into MLLM, then incorporating with RGB-D images and mask of the current frame to produce expanded expressions $\mathcal{E}_{o_i t}$ that capture subtle states and interactions via precise prompting:

$$\mathcal{E}_{o_i t} = \Psi_{MLLM}(e_{o_i t}, I_t, D_t, \mathcal{M}_{o_i t}, \mathcal{P}), \tag{2}$$

where $D_t$, $\mathcal{P}$ denote depth map and text prompt. Detailed prompts can be found in Appendix. D. Then, we employ a frozen LLM (Lee et al., 2020) to enrich language embeddings from $f_{o_i t}^{\ell} \in \mathbb{R}^{d_e} = \Psi_{LLM}(\mathcal{E}_{o_i t})$, which subsequently supervise the Gaussian language representations via the hybrid injection strategy.

### 3.2.2 DYNAMIC RETRIEVABLE RECONSTRUCTION WITH UNIFIED VISION-LANGUAGE FEATURES

Beyond photometric and geometric fidelity, we aim to learn a dynamic 4D feature representation that supports scene-level instrument retrieval. To this end, we adopt a flexible dynamic Gaussian modeling strategy (Yang et al., 2024b; Lin et al., 2024), leveraging a dynamic temporal deformation model to capture the evolution of Gaussian primitives across time. The trajectory of each Gaussian primitive is parametrically represented. For a Gaussian $g^i$ at reference time $t_0$ (set to the first frame), its basic states $\Theta(t) := (\mu_t, r_t, s_t)$ at any time $t$ are modeled as a linear combination of $J$ learnable Gaussian basis functions:

$$\Theta(t) = \Theta_0 + \sum_{j=1}^{J} w_j \phi_j(t; \theta_j, \sigma_j), \quad \phi_j(t; \theta_j, \sigma_j) = \exp\left(-\frac{(t - \theta_j)^2}{2\sigma_j^2}\right), \tag{3}$$

where $\Theta_0$ is the initial state, $\mu_t$, $r_t$, $s_t$ are position, rotation and scaling factor, respectively; $w_j$, $\theta_j$, $\sigma_j$ are learnable parameters. This formulation enables a smooth and efficient modeling of Gaussian point movements in dynamic scenarios, where the basis functions ensure locality and smoothness while capturing high-frequency details like instrument vibrations and tissue pulsations.

Furthermore, we associate the Gaussian primitives that contribute to each pixel with spatiotemporally consistent semantics. Specifically, we attach to each Gaussian $g^i$ a unified latent vector $\mathbf{z}_i \in \mathbb{R}^{d_v + d_l}$ partitioned as $\mathbf{z}_i = [\mathbf{f}_i^v, \mathbf{f}_i^l]$, where $\mathbf{f}_i^v \in \mathbb{R}^{d_v}$, $\mathbf{f}_i^l \in \mathbb{R}^{d_l}$ denote the visual and the language subspaces respectively. These

subspaces are optimized jointly and interact within the feed-forward rasterization process, establishing $\mathbf{z}_i$ as a synergistic multimodal carrier rather than independent components. Formally, as in Eq.1, the 2D visual embedding map $\hat{\mathcal{F}}^v$ and language embedding map $\hat{\mathcal{F}}^l$ at pixel $p$ at a given time point are expressed as:

$$[\hat{\mathcal{F}}^v(p), \hat{\mathcal{F}}^l(p)] = \sum_{i=1}^{n} \mathbf{z}_i o_i G_i \prod_{j=1}^{i-1} (1 - o_j G_j), \tag{4}$$

where $\hat{\mathcal{F}}^v \in \mathbb{R}^{d_v \times H \times W}$ and $\hat{\mathcal{F}}^l \in \mathbb{R}^{d_l \times H \times W}$ are jointly learned, with the dual-branch learning strategy described below for holistic Gaussian representations.

**Dynamic visual representation learning with spatial structural prior**. We distill class-agnostic structural features $\mathcal{F}^v \in \mathbb{R}^{ds \times H \times W}$ from SAM (Kirillov et al., 2023), providing boundary-aware priors to guide visual representation learning. This ensures that Gaussians capture both surface appearance and structural details, allowing for precise localization of tools and tissues even under occlusion or deformations, and maintaining spatial integrity during interactions. Moreover, to achieve spatial-temporal consistent reconstructions, we introduce a temporal modulation module $\Gamma$ through lightweight MLPs. Then, we align the visual representation $\mathbf{f}_i^v$ with color features $c^i$ over time, ensuring smooth rendering with reduced artifacts. The time-modulated dynamic visual representation $\mathbf{f}_i^v$ at time $t$ is computed as:

$$\mathbf{f}_i^v(t) = \mathbf{f}_i^v(t) + \Gamma_v \left( c_i(t) \oplus \mathbf{f}_i^v \oplus \mathrm{PE}(t) \right), \tag{5}$$

where $\oplus$ denotes concatenation and $\mathrm{PE}(\cdot)$ is position encoding. The $\mathbf{f}_i^v$ is concatenated with the dynamic language representations $\mathbf{f}_i^l$ (detailed later) to obtain the visual feature map $\hat{\mathcal{F}}^v$, as in Eq.4. For a pixel $p$, we apply L1 loss for alignment: $\mathcal{L}_v = ||\hat{\mathcal{F}}^v(p) - \mathcal{F}^v(p)||_1$. Meanwhile, the color map $\hat{C}$ at pixel $p$ is:

$$\hat{C}(p) = \sum_{i=1}^{n} \mathbf{c}_i^*(t) o_i G_i \prod_{j=1}^{i-1} (1 - o_j G_j), \quad \mathbf{c}_i^*(t) = c_i(t) + \Gamma_v \left( c_i(t) \oplus \mathbf{f}_i^v \oplus \mathrm{PE}(t) \right), \tag{6}$$

where $\mathbf{c}_i^*$ is the color feature of Gaussian $g^i$ after temporal modulation and structural feature augmentation.

**Dynamic language representation learning with hybrid semantic injection**. In dynamic surgical scenes, embedding language semantics into Gaussian primitives is crucial for semantic reconstruction and free-form retrieval. Current solutions rasterize a Gaussian language field via volume rendering for efficient alignment but lack consistency, as a single Gaussian may map to multiple pixels (Fig. 4). Others enforce consistency through explicit 3D–2D projection, yet they rely on unreliable depth estimated in endoscopy with noisy inputs. To address these, we propose a hybrid semantic injection scheme that combines volumetric rendering with weighted semantic registration for coarse alignment and spatial-temporal consistency. Inspired by Jun-Seong et al. (2025), our approach realizes depth-free semantic injection by selecting top-K contributing Gaussians per pixel and assigning semantics weighted by their contributions. The temporal-modulated language features at any time $t$ are first computed by:

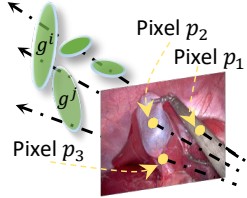

Figure 4: Ambiguity in semantics embedding.

$$\mathbf{f}_i^l(t) = \mathbf{f}_i^l(t) + \Gamma_l \left( \mathbf{f}_i^l \oplus \mathrm{PE}(t) \right). \tag{7}$$

Combined with the dynamic visual embedding $\mathbf{f}_i^v$, the language semantic map $\hat{\mathcal{F}}^l$ is computed by Eq.4 and aligned with the enriched expression feature map $\mathcal{F}^l \in \mathbb{R}^{d_e \times H \times W}$ using $\mathcal{L}_{render}^l = ||\hat{\mathcal{F}}^l(p) - \mathcal{F}^l(p)||_2^2$. For weighted language semantic injection, the contribution weights $w_i$ for Gaussian $g^i$ at pixel $p$ is computed with Eq.1: $w_i = o_j G_j \cdot \prod_{j=1}^{i-1} (1 - o_j G_j)$. We select the top-$K$ Gaussians by sorting these weights, and the semantics for pixel $p$ is aggregated by: $\hat{\mathcal{F}}^l(p) = \sum_{k=1}^{K} w_k \mathbf{f}_k^l / \sum_{k=1}^{K} w_k$. We further align it with the corresponding semantics by cosine similarity, *i.e.*, $\mathcal{L}_{align}^l = 1 - cossim(\hat{\mathcal{F}}^l(p), \mathcal{F}^l(p))$. To maintain

Table 2: Quantitative comparisons of 4D surgical scene reconstruction on three datasets.

| Methods | StereoLung15K | | | | StereoMIS | | | | EndoNerf | | | |
|---|---|---|---|---|---|---|---|---|---|---|---|---|
| | PSNR↑ | SSIM↑ | LPIPS↓ | RMSE↓ | PSNR↑ | SSIM↑ | LPIPS↓ | RMSE↓ | PSNR↑ | SSIM↑ | LPIPS↓ | RMSE↓ |
| LGS (Liu et al., 2024) | 22.165 | 68.073 | 0.564 | - | 26.765 | 73.629 | 0.333 | - | 30.185 | 88.483 | 0.221 | - |
| Endo4DGS (Huang et al., 2024) | 21.778 | 67.846 | 0.537 | 7.082 | 26.498 | 73.714 | 0.324 | 6.363 | 30.366 | 90.619 | 0.157 | 5.573 |
| ForPlane32k (Yang et al., 2024a) | 20.933 | 66.121 | 0.455 | 9.106 | 26.723 | 70.373 | 0.253 | 8.255 | 29.096 | 87.535 | 0.179 | 5.728 |
| Deform3DGS (Yang et al., 2024b) | 22.688 | 68.866 | 0.454 | 6.173 | 27.323 | 77.822 | 0.249 | 4.809 | 32.172 | 92.509 | 0.106 | 4.747 |
| SurgTPGS (Huang et al., 2025) | 20.913 | 66.996 | 0.589 | 9.707 | 26.431 | 75.301 | 0.284 | 8.917 | 30.271 | 90.844 | 0.137 | 9.825 |
| EndoGaussian (Liu et al., 2025) | 22.642 | 68.227 | 0.508 | 5.987 | 26.768 | 73.176 | 0.329 | **3.364** | 30.013 | 88.448 | 0.216 | 4.543 |
| EHSurGS (Shan et al., 2025) | 23.401 | 71.437 | 0.483 | 5.964 | 27.730 | 77.980 | 0.256 | 3.823 | 31.90 | 91.888 | 0.131 | 5.385 |
| EndoRDGS (Gao et al., 2025) | 23.868 | 72.161 | 0.443 | 7.67 | 28.443 | 82.15 | 0.176 | 5.600 | 32.744 | 93.0 | 0.088 | 4.87 |
| 4DGS (Wu et al., 2024a) | 23.231 | 70.496 | 0.493 | 6.344 | 27.221 | 76.222 | 0.28 | 4.108 | 31.132 | 91.725 | 0.128 | 4.78 |
| DGD (Labe et al., 2024) | 23.583 | 72.264 | 0.438 | - | 26.614 | 76.234 | 0.285 | - | 31.868 | 92.728 | 0.152 | - |
| 4DLangSplat (Li et al., 2025b) | 23.307 | 70.458 | 0.499 | 5.908 | 27.076 | 76.126 | 0.289 | 4.138 | 31.064 | 91.561 | 0.134 | 4.772 |
| *SurgRe4DGS* (ours) | **24.524** | **73.502** | **0.372** | **5.891** | **28.692** | **82.617** | **0.146** | 3.605 | **32.808** | **93.105** | **0.083** | **4.337** |

semantic distinctiveness across instruments, we also incorporate contrastive constraints to increase inter-instrument feature distances, described as:

$$\mathcal{L}_{\text{cons}}^l = -\frac{1}{|\{i : |\mathcal{Q}_i^+| > 0\}|} \sum_{i:\,|\mathcal{Q}_i^+|>0} \frac{1}{|\mathcal{Q}_i^+|} \sum_{j\in\mathcal{Q}_i^+} w_i w_j \, \log \frac{\exp(s_{ij})}{\exp(s_{ij}) + \sum_{h\in\mathcal{Q}_i^-} \exp(s_{ih})}, \quad s_{uv} = \frac{\mathbf{f}_u^l \cdot \mathbf{f}_v^l}{\tau}, \quad (8)$$

where $\mathcal{Q}_i^+$ and $\mathcal{Q}_i^-$ denote the positive and negative sets for Gaussian point $i$, respectively, and $\tau$ is the temperature coefficient. Totally, we have $\mathcal{L}_l = \mathcal{L}_{render}^l + \mathcal{L}_{align}^l + \mathcal{L}_{cons}^l$.

**Overall Optimization**. Finally, our *SurgRe4DGS* is optimized in a multi-objective manner to build a robust 4D vision-language representation in dynamic surgical scenes. The overall loss is defined as:

$$\mathcal{L} = \mathcal{L}_{color} + \mathcal{L}_{depth} + \lambda_v \mathcal{L}_v + \lambda_l \mathcal{L}_l, \quad (9)$$

where $\mathcal{L}_{color} = ||\hat{C}(p) - C(p)||_1$, $\mathcal{L}_{depth} = ||\hat{D}(p) - D(p)||_1$ denote the photometric losses; $C$ and $D$ are the reference RGB-D images; $\lambda_v$ and $\lambda_l$ are the hyperparameters.

## 4 EXPERIMENTS

**Datasets and Metrics.** We evaluate *SurgRe4DGS* with other approaches on *StereoLung15K* and two public datasets, EndoNerf (Wang et al., 2022) and StereoMIS (Hayoz et al., 2023), for 4D reconstruction and open-vocabulary instrument retrieval. To enable language-driven retrieval, the instrument semantics of two public datasets are annotated in the same way as in Sec.2. Following the standard evaluation (Gao et al., 2025; Shan et al., 2025; Yang et al., 2024b; Wang et al., 2022), we use the same sequences and split training/testing sets at a 7:1 ratio per scene for a fair comparison. We compare against state-of-the-art methods, and metrics include PSNR, SSIM(%), LPIPS for photometric quality, RMSE(mm) for geometry quality, and mIoU, mDice, mRecall for retrieval. Detailed implementation is available in Appendix. D.

### 4.1 COMPARISONS FOR 4D SURGICAL SCENE RECONSTRUCTION

Tab.2 showcases quantitative comparison results on the *StereoLung15K* test set, where our method achieves the best performance in photometric and geometry reconstruction. *StereoLung15K* captures challenges of dynamic surgical scenes, including intricate tissue deformations and instrument motion, which existing methods struggle to handle. In contrast, our method effectively models these dynamics, achieving 24.524 PSNR and 73.502 SSIM, significantly outperforming existing approaches, like EndoRDGS and 4DLangSplat. The superiority of our framework also shows in public benchmarks, with attaining 28.692

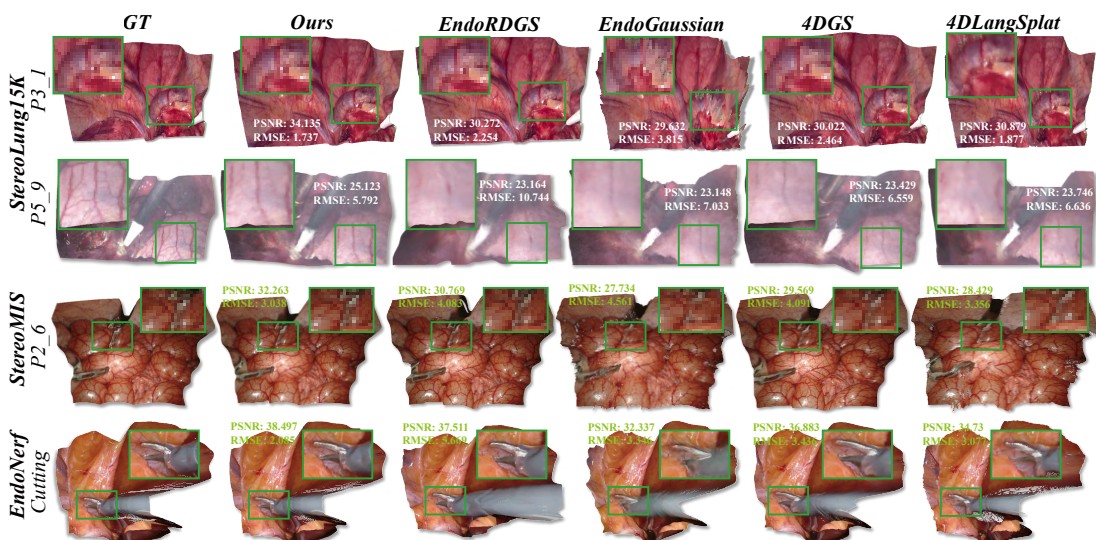

Figure 5: Qualitative comparisons of 4D surgical scene reconstruction.

Table 3: Quantitative comparisons of 4D open-vocabulary instrument retrieval on three datasets.

| Methods | StereoLung15K | | | StereoMIS | | | EndoNerf | | |
|---|---|---|---|---|---|---|---|---|---|
| | mIoU↑ | mDice↑ | mRecall↑ | mIoU↑ | mDice↑ | mRecall↑ | mIoU↑ | mDice↑ | mRecall↑ |
| SurgTPGS (Huang et al., 2025) | 43.758 | 51.798 | 0.596 | 42.972 | 48.152 | 0.549 | 49.612 | 57.210 | 0.529 |
| DGD (Labe et al., 2024) | 49.884 | 58.359 | 0.608 | 52.584 | 58.745 | 0.632 | 53.055 | **59.267** | 0.615 |
| 4DLangSplat (Li et al., 2025b) | 43.914 | 51.880 | 0.565 | 41.775 | 49.599 | 0.483 | 45.379 | 52.923 | 0.530 |
| *SurgRe4DGS* (ours) | **57.907** | **65.908** | **0.726** | **59.767** | **64.955** | **0.708** | **53.558** | 59.130 | **0.642** |

PSNR on StereoMIS and 32.808 on EndoNerf, exceeding 4DGS by 1.47 and 1.68, respectively, and also surpassing SOTA methods like EndoRDGS. Fig.5 provides qualitative comparisons across the three benchmarks, showing *SurgRe4DGS* excels even amid complex anatomical structures and severe smoke disturbances ($P5\_9$). These results validate our framework's effectiveness, with more details in Appendix. F.

## 4.2 COMPARISONS FOR OPEN-VOCABULARY INSTRUMENT RETRIEVAL

Tab.3 presents quantitative comparisons of language-driven tool queries on the test sets of three benchmarks, where *SurgRe4DGS* achieves state-of-the-art performance across all datasets, notably outperforming DGD and 4DLangSplat by 8% and 14% in mIoU on the *StereoLung15K*. We further display the qualitative comparisons of tool localization with different query strategies, *e.g.*, '*type, action, position*' and '*type, action*' on the StereoMIS and *StereoLung15K* datasets. As shown in Fig.6, *SurgRe4DGS* outperforms existing approaches and can locate the target accurately, even in the case of multiple instrument instances where existing methods fail to handle. The similarity map also displays Gaussian activation distributions, with red areas indicating activated contributing Gaussians. These results highlight the efficacy of our hybrid semantics injection strategy. More detailed results are displayed in Appendix. F.

## 4.3 ABLATION STUDY

**Effect of spatial structural prior.** We study the impact of the spatial structural prior, distilled from the segmentation foundation model, on reconstruction and retrieval performance. We compare variants with

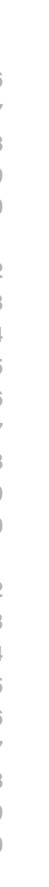

Query: "Prograsp forceps is idle on the left"    Query: "Ultrasonic scalpel is cutting tissue"

Figure 6: Qualitative comparisons of 4D surgical instrument retrieval with varying query types.

Table 4: Effect of spatial structural prior.

| Methods | StereoLung15K* | | | StereoMIS | | |
|---|---|---|---|---|---|---|
| | PSNR↑ | RMSE↓ | mIoU↑ | PSNR↑ | RMSE↓ | mIoU↑ |
| W/o | 26.131 | 5.582 | 55.248 | 27.678 | 3.751 | 56.273 |
| W/ | **26.724** | **5.354** | **56.615** | **28.692** | **3.606** | **59.767** |

Table 5: Effects of hybrid semantics injection strategy.

| $\mathcal{L}_l$ | | | StereoLung15K* | | | StereoMIS | | |
|---|---|---|---|---|---|---|---|---|
| $\mathcal{L}_{render}^l$ | $\mathcal{L}_{align}^l$ | $\mathcal{L}_{cons}^l$ | PSNR↑ | RMSE↓ | mIoU↑ | PSNR↑ | RMSE↓ | mIoU↑ |
| ✓ | | | 26.665 | 5.449 | 55.143 | 28.677 | 3.674 | 56.299 |
| | ✓ | | 26.657 | 5.434 | 48.938 | 28.373 | 4.064 | 44.635 |
| ✓ | ✓ | | 26.699 | 5.428 | 55.702 | 28.683 | 3.622 | 57.663 |
| ✓ | ✓ | ✓ | **26.724** | **5.354** | **56.615** | **28.692** | **3.606** | **59.767** |

(W/) and without (W/o) this prior by ablating visual representation learning in Sec.3.2.2. As shown in Tab. 4, incorporating the prior consistently improves PSNR by 0.593 and 1.014, reduces RMSE by 0.228 and 0.145, and boosts mIoU by 1.367 and 3.494, respectively. These improvements highlight the prior's role in enhancing boundary delineation and structural robustness under intricate surgical conditions.

**Effects of hybrid semantics injection strategy.** To verify the efficacy of our hybrid semantics injection strategy, we progressively ablate the key constraint terms, as shown in Tab.5. Relying solely on the volume rendering or 3D-2D weighted semantics aggregation degrades the performance of both 4D reconstruction and language-driven tool retrieval (particularly the latter). In contrast, our proposed hybrid injection strategy yields substantial improvements, and with the cross-instance semantics constraint, the full model attains optimal performance in 4D retrievable scene reconstruction. This confirms the strategy's efficacy for superior reconstruction fidelity and open-vocabulary retrieval accuracy in complex surgical scenes.

**Impact of top-K selection.** We evaluate the top-K parameter in our semantic injection strategy by varying K across five values, as shown in Fig.7. On StereoMIS, PSNR remains around 28.7, while RMSE reaches its minimum and mIoU peaks at K=20 before slight decreases. These trends indicate that increasing K helps up to a point, after which noise accumulation yields diminishing returns. We select K=20 as the default for balanced fidelity and efficiency. More ablation studies are presented in Appendix. E.

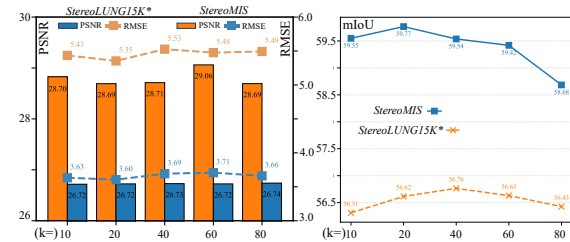

Figure 7: Impact of top-K selection.

## 5 CONCLUSION

In this work, we present *SurgRe4DGS*, a novel framework that advances 4D surgical scene reconstruction by integrating open-vocabulary capabilities through unified vision-language Gaussian representations, dual-branch integration, and hybrid language injection. This design overcomes limitations in reconstructing dynamic surgical environments while achieving robust instrument retrieval, facilitating intraoperative perception and decision-making. The released *StereoLung15K* further addresses data scarcity, offering a high-quality benchmark for reconstruction and retrieval tasks in intraoperative surgical scenes. Quantitative and qualitative results demonstrate that our method achieves superior performance and highlight the potential of both our method and dataset for real-world clinical deployment.

## 6 ETHICAL STATEMENT

This study complies with the ICLR Code of Ethics, with all procedures involving human subjects adhering to institutional and national ethical standards and approved by the Institutional Review Board (IRB) of collaborating hospitals. To safeguard patient privacy and confidentiality, all surgical videos were anonymized by removing identifiable information, such as patient details, faces, or timestamps, ensuring fully de-identified data for research purposes. No conflicts of interest exist, and the work aims to advance beneficial applications in surgical AI without introducing risks of bias, harm, or privacy violations.

## 7 REPRODUCIBILITY STATEMENT

All key technical details of *SurgRe4DGS* are provided in the main paper and Appendix, including architectures, optimization strategies, hyperparameters, training procedures, and quantitative/qualitative results. The *StereoLung15K* dataset is fully described in Sec.2 and Appendix, with annotation protocols, statistics, and detailed benchmarking results. This dataset will be publicly released upon acceptance, including raw video frames, masks, and text semantics. All experiments were conducted using standard libraries (*e.g.*, Pytorch), with random seeds fixed for consistency.

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

## A  APPENDIX

Our appendix includes the following contents:

- Section B: **Related Works**. Providing more recent works related to this work.
- Section C: *StereoLung15K* **Dataset**. Showing more detailed information and statistics about the proposed dataset.
- Section D: **Implementation Details**. Giving more exhaustive implementation details and experimental settings.
- Section E: **More Ablation Studies**. Reporting more ablation studies.
- Section F: **Additional Results**. Presenting additional scene-wise comparison results and visualization results.
- Section G: **Broader Applications, Limitations and Future Work**. Discussing potential broader application scenarios of *SurgRe4DGS* and its limitations, future work.
- Section H: **The Use of Large Language Model**.
- Section I: **Notation Table for Key Parameters and Symbols**.

## B  RELATED WORKS

**4D Representations for Dynamic Reconstruction.** Recent advances in 4D representations focus on capturing temporal dynamics in scene reconstruction, extending static 3D methods to handle motion and deformations. In natural scenes, neural radiance fields (NeRF) (Mildenhall et al., 2021) have been extended to dynamic settings (Du et al., 2021; Park et al., 2021; Weng et al., 2022; Cao & Johnson, 2023; Fridovich-Keil et al., 2023), such as Dynamic NeRF (Pumarola et al., 2021) and TiNeuVox (Fang et al., 2022), which model time-varying densities and colors using deformable fields. Recently, 3D Gaussian (Kerbl et al., 2023) has emerged as a widely used explicit representation for 3D scenes (Feng et al., 2025; Lin et al., 2025), and various extensions have been proposed for dynamic 4D scene representations (Yang et al., 2024c; Duan et al., 2024; Stearns et al., 2024; Yang et al., 2024d; Lei et al., 2025). For instance, 4D-GS (Wu et al., 2024a) uses time-dependent Gaussians with deformation networks to achieve real-time rendering of dynamic scenes. 4D-fly (Wu et al., 2025) introduces a streaming 4D reconstruction that injects priors into explicit Gaussians, anchors temporal propagation, and optimizes for visual and motion fidelity. These methods excel in general environments but often assume clean, structured motion, limiting their adaptability to irregular dynamics.

When transitioning to surgical domains, 4D reconstruction is crucial for capturing deformable anatomy (*e.g.*, organs, tissues) from surgical video, where scenes involve non-rigid tissue deformations and instrument interactions (Wang et al., 2024a). NeRF-based methods have also been attempted for surgery (Wang et al., 2022; Zha et al., 2023; Yang et al., 2024a). However, they require lengthy training times per scene and low rendering speed, which is impractical for intraoperative use. 3D Gaussian has recently been introduced to this domain to tackle these issues (Liu et al., 2024; Huang et al., 2024; Liu et al., 2025; Zheng et al., 2025; Shan et al., 2025). Deform3DGS (Yang et al., 2024b) introduces flexible deformation modeling with point cloud initialization for fast surgical scene reconstruction, addressing sparse textures and occlusions. EndoRDGS (Gao et al., 2025) proposes periodic modulated Gaussian functions and a Biplane module to enhance local tissue deformation representation and enable stable spatiotemporal adjustments, achieving superior performance in real-time endoscopic surgical scene reconstruction. These works validate that 4D Gaussian representations can capture fine anatomical motions and outperform prior surgical reconstruction methods in both speed and realism. *However, existing 4D reconstruction methods in surgical scenarios largely focus on geometry and appearance, without leveraging high-level semantic understanding of the scene. This gap motivates the integration of language and semantic information into dynamic reconstructions.*

**Language-embedded Gaussian Splatting.** Bridging 3D reconstruction with semantic and language understanding has become an important research direction. Early progress was made by injecting language or visual-semantic features into neural fields (Kobayashi et al., 2022; Tschernezki et al., 2022; Wang et al., 2024b), such as LERF (Kerr et al., 2023), which distills CLIP features into neural radiance fields to enable text-based 3D queries. However, NeRF-based pipelines are slow and implicit, making it difficult to retrieve or edit fine-grained semantic content. With the advent of Gaussian splatting, several works leverage its efficiency to build language-embedded 3D representations (Shi et al., 2024; Ye et al., 2024; Halacheva et al., 2025; Peng et al., 2024). Feature3DGS (Zhou et al., 2024) pioneers augmenting 3D-GS with high-dimensional features distilled from 2D foundation models (like SAM and CLIP), effectively creating a semantic feature field over the Gaussian points. Similarly, LangSplat (Qin et al., 2024) constructs a 3D language field by grounding each Gaussian with features from vision-language models, which uses Segment-Anything Model (SAM) to generate precise masks and associates each Gaussian with a CLIP embedding, achieving accurate open-vocabulary object labeling in the 3D scene. Concurrently, OpenGaussian (Wu et al., 2024b) proposes to provide efficient open-vocabulary 3D understanding at the point level, demonstrating that one can attach semantic descriptors to Gaussians for tasks like labeling and retrieval. These methods report that explicit Gaussian-based representations, enhanced with language features, can attain competitive semantic precision while running in real time. Building on the above advances, more recent studies push the envelope of semantic Gaussian splatting. Works like Dr.Splat (Jun-Seong et al., 2025) and InstanceGaussian (Li et al., 2025a) have been introduced to incorporate instance-level knowledge and more complex semantics into Gaussian fields, *i.e.*, by associating each Gaussian with object instance IDs or enabling instance-wise manipulation of the scene.

In parallel, researchers have extended language-embedded Gaussians to 4D dynamic scenes (Fiebelman et al., 2025; Zhou et al., 2025). DGD (Labe et al., 2024) introduces a unified 3D representation that jointly optimizes color and semantic attributes for dynamic 3D scene reconstruction, enabling dense semantic object tracking and fast rendering. 4D LangSplat (Li et al., 2025b) learns both time-invariant and time-varying language embeddings on deformable Gaussians, using multimodal large language models to generate detailed captions for each object over time, achieving open-vocabulary queries in videos. In the surgical realm, integrating language and semantic supervision into 3D reconstruction is still nascent. To our knowledge, the first attempt is SurgTPGS (Huang et al., 2025), which incorporates language guidance for recognizing instruments and structures. However, it remains limited in scope, focusing on task-specific recognition and lacking support for open-vocabulary querying or continuous temporal changes. *Overall, while general-domain research has started to merge 4D reconstruction with language semantics, the surgical domain is still lagging behind. The real-time, high-fidelity reconstruction required for surgical environments, coupled with the difficulty of modeling the non-rigid, often occluded anatomy, makes 4D language-conditioned surgical scene reconstruction a particularly difficult problem. These challenges highlight the gaps in current methods and emphasize the need for more robust approaches capable of handling both dynamic surgical scenes and language-driven semantic understanding. Our work aims to address these gaps by providing both the necessary datasets and innovative methods for dynamic, language-guided 4D surgical scene reconstruction.*

## C *StereoLung15K* DATASET

### C.1 DATA EXHIBITION

Fig. 8 displays some representative samples of our *StereoLung15K* dataset. These samples highlight the dataset's realism and complexity in real-world surgical scenarios, including blood flow that obscures visibility, multiple instrument instances interacting simultaneously, smoke from cauterization tools, and occlusions caused by overlapping devices or tissues. These features pose significant challenges for 4D reconstruction and open-vocabulary instrument retrieval tasks, making the dataset a valuable benchmark for evaluating the robustness of models in dynamic, cluttered environments.

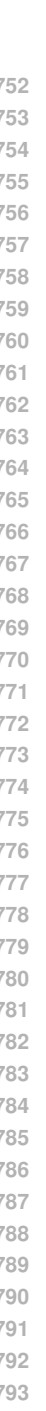
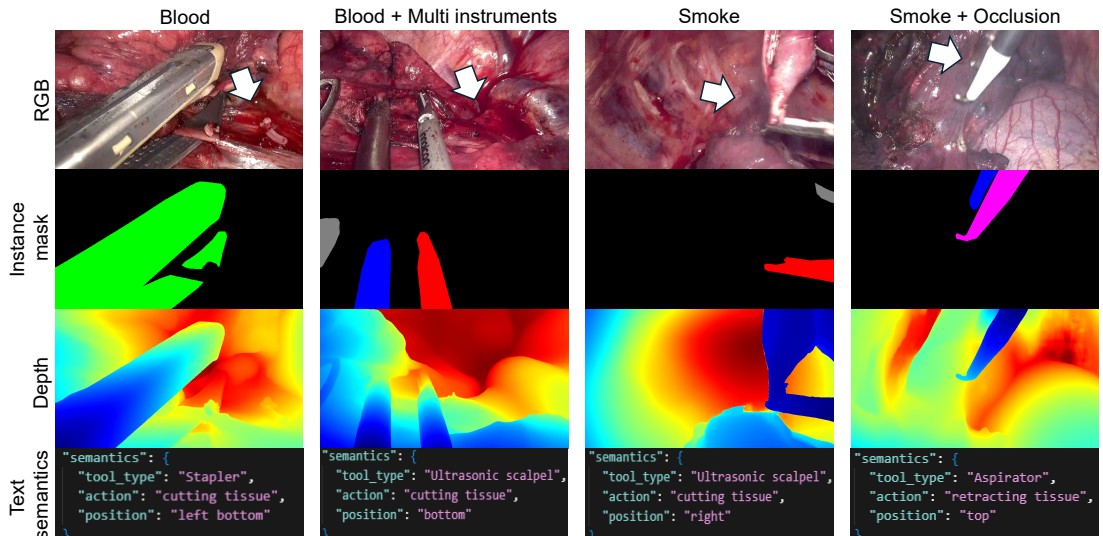

Figure 8: Representative examples of our *StereoLung15K* dataset.

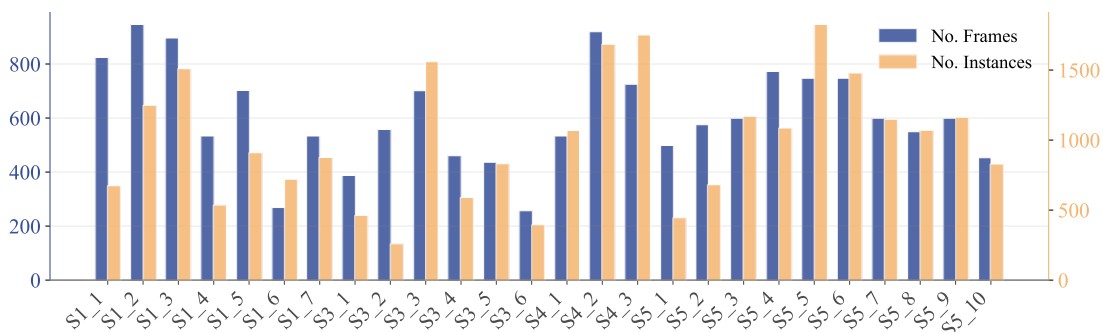

Figure 9: Statistics of frames and instances for each sequence in *StereoLung15K* dataset.

## C.2 MORE STATISTICS

We further provide additional statistics for our proposed *StereoLung15K*. Fig. 9 provides the detailed statistics on frames and instances per sequence. Fig. 10 and Fig. 11 illustrate the cross-attribute distributions, showing the interrelationship between instruments and actions, as well as instruments and positions. The action distribution reveals frequent pairings, such as "Aspirator" with "retracting tissue" or "Oval grasper" with "grasping tissue", reflecting common surgical workflows where instruments are linked to specific tasks like manipulation or coagulation. Similarly, the positional preferences shown in Fig. 11 further highlight spatial dynamics in surgical endoscopic views.

Table 6: Percentage distribution of instrument instances per frame between CholecT50 and *StereoLung15K*.

| Dataset | 0 | 1 | 2 | 3 |
|---|---|---|---|---|
| CholecT50 Nwoye et al. (2022) | 10.970% | 35.18% | 47.1% | 6.75% |
| *StereoLung15K* | 9.19% | 30.09% | 48.21% | 12.5% |

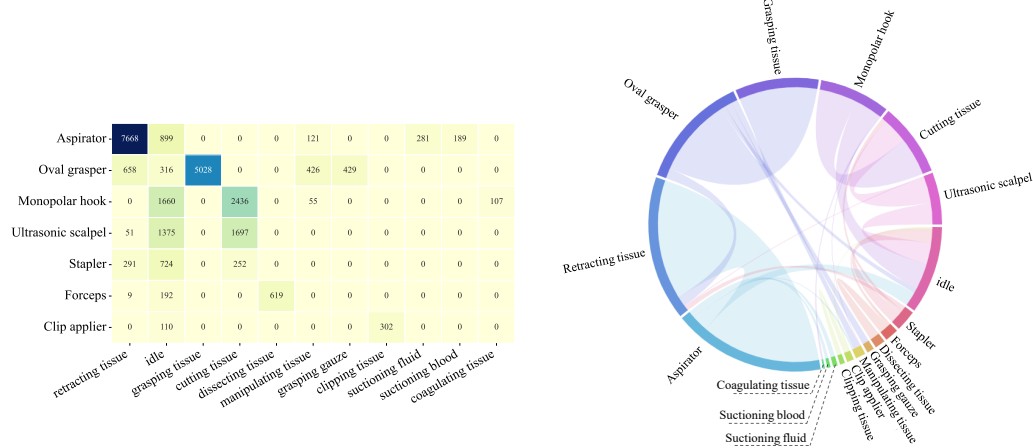

Figure 10: **Attribute distribution**. Left: Co-attributes distribution of *Instrument-Action* over *StereoLung15K*. The number in each grid indicates the total number of instances. Right: the interrelationships among multiple instrument instances and actions.

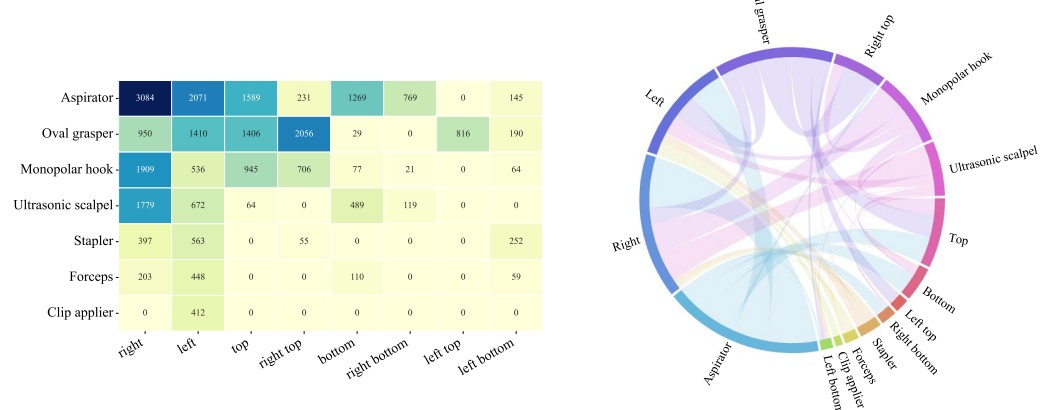

Figure 11: **Attribute distribution**. Left: Co-attributes distribution of *Instrument-Position* over *StereoLung15K*. The number in each grid indicates the total number of instances. Right: the interrelationships among multiple instrument instances and positions.

In Tab. 6, we compare our *StereoLung15K* with the most popular instrument triplet detection dataset in terms of distribution of instrument instances per frame. Our dataset exhibits a higher proportion of frames with 3 instruments (12.5% vs. 6.75%), indicating better scene complexity and multi-tool interactions, while maintaining similar patterns for 0-2 instances. This underscores *StereoLung15K*'s value for benchmarking models in realistic, crowded surgical environments.

# D  IMPLEMENTATION DETAILS

All experiments are conducted on a single NVIDIA 3090 GPU using the PyTorch framework. For all baselines, we use the official public code and train with the hyperparameters reported by the authors. For the StereoMIS (Hayoz et al., 2023) public datasets evaluation, we selected four scenes in line with existing methods for a fair comparison, *i.e.*, `P2_6_9100_9480`, `P3_9100_9950`, `P3_11000_11400`, `P3_11200_11525`. For the EndoNerf (Wang et al., 2022) dataset, we selected 5 scenes, specifically: `cutting_tissues_twice`, `pulling_soft_tissues`, `pushing`, `thin`, `traction`. We use the FoundationStereo (Wen et al., 2025) to generate the depth map for the left-view images. For ablation studies, we use the StereoMIS dataset and three scenes from *StereoLung15K* (`P4_1`, `P4_2`, `P4_3`).

**Feature Dimensions.** We compress the high-dimensional distilled class-agnostic structural features $\mathcal{F}^v \in \mathbb{R}^{ds \times H \times W}$ ($ds$=256) and language features $\mathcal{F}^l \in \mathbb{R}^{d_e \times H \times W}$ ($d_e$=768) to accelerate the training process following Feature3DGS (Zhou et al., 2024) and 4DLangSplat (Li et al., 2025b). Specifically, for the feature $\mathcal{F}^v$, we utilize a learnable $1 \times 1$ convolution layer during GS training, facilitating channel-wise communication between the high-dimensional rendered feature and Gaussian primitives, achieving 4x rendering speed (*i.e.*, compressed $ds$=64) simultaneously. For feature $\mathcal{F}^l$, we utilize an autoencoder to compress the original feature to 6 dimensions. The autoencoder consists of tiny MLPs and is optimized with L2 loss.

**Training Details.** We employ a progressive training strategy to gradually refine our *SurgRe4DGS* framework for dynamic, language-driven 4D surgical scenes reconstruction. In the first stage, we integrate the spatial priors into Gaussian model training to establish a holistic representation of the basic visual-geometry properties. In the second stage, we align the Gaussians with language embeddings, injecting them with semantic understanding of the scene. Finally, we incorporate the cross-instance contrastive constraint to enhance the uniqueness and distinctiveness of semantic features for instruments within the scene. For all datasets, the iterations for the three stages are set to 10,000, 8,000, and 2,000. The learning rates for both the deformable network and the temporal-modulation network are set to $1.6 \times 10^{-4}$. The number of learnable Gaussian basis functions in Eq. 3 is set to 17. Other training parameters remain consistent with 4DGS (Wu et al., 2024a).

**Instrument-Wise Semantics Enrichment via MLLM.** We leverage the Qwen2.5-VL-7B-Instruct model Bai et al. (2025) as the backbone MLLM to generate fine-grained, instrument-wise semantics descriptions. The detailed prompt is provided in Tab.7. Fig.12 further presents examples of enriched descriptions.

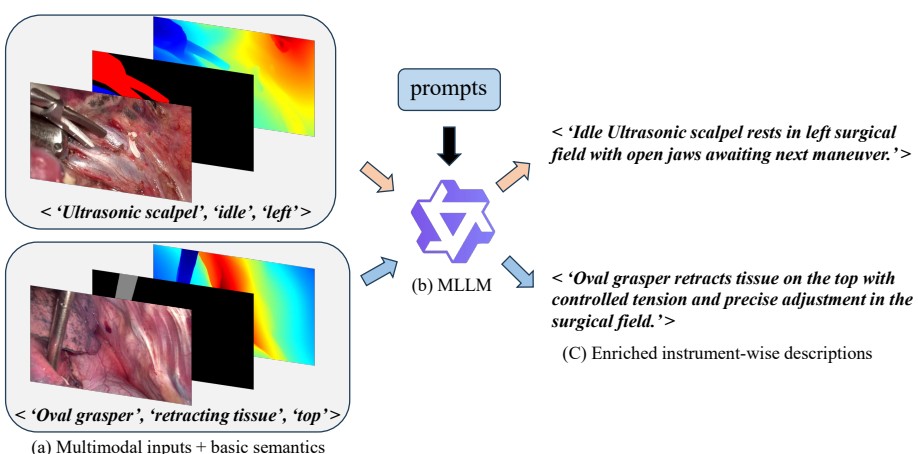

Figure 12: Examples of enriched instrument-wise description in our *StereoLung15K* dataset.

Table 7: Details of Text Prompts.

---

**Your task is to ENRICH the basic expression while MAXIMIZING embedding distinguishability between different surgical instrument states and actions. The enriched description will be used with text encoders to create embeddings for 4D Gaussian scene retrieval.**

**CRITICAL REQUIREMENT**: Avoid template based descriptions that make different instruments or actions too similar. Instead, create distinctive descriptions that emphasize unique characteristics of each instrument action combination.

**Embedding Optimization Strategy**:
- **Action First Approach**: Lead with distinctive action verbs that create semantic separation.
- **Instrument Specific Vocabulary**: Use different descriptor patterns for each instrument type.
- **Spatial Diversity**: Vary spatial description styles to avoid repetitive patterns.
- **Context Variation**: Include different environmental or visual contexts for similar actions.
- **Semantic Layering**: Combine multiple unique semantic dimensions without uniform templating.

**Key Requirements**:
- PRESERVE exact instrument names from basic expression.
- Lead with the most distinctive element which can be the action the context or a unique feature.
- Use different sentence structures for different instrument action combinations.
- Include two to three unique semantic dimensions per description.

**CRITICAL**: Each action type must use DIFFERENT sentence structures and vocabulary patterns to maximize embedding distinguishability. Avoid any repetitive phrasing across similar actions.

**Format**: Provide ONLY the enriched expression as a single sentence. Each description must have a UNIQUE semantic pattern to maximize embedding distinguishability.

---

**Open-Vocabulary 4D Instrument Retrieval.** Conventional evaluations of open-vocabulary models often rely on predefined category names for quantitative assessment, which fail to reflect their ability to handle diverse natural language queries in real-world scenarios. This limitation is especially evident in surgical contexts, where instruments are described not only by type but also by dynamic attributes like actions and positions (*e.g.*, "forceps dissecting the tissue") to reflect intraoperative needs. To better evaluate model flexibility and robustness, we propose a comprehensive protocol that uses four query types derived from enriched annotations: 1) *instrument type alone* (*e.g.*, "forceps"); 2) *type + action* (*e.g.*, "Ultrasonic scalpel is retracting tissue"); 3) *type + position* (*e.g.*, "Aspirator is on the left"); and 4) *type + action + position* (*e.g.*, "Monopolar hook is idle on the left"). This design mirrors clinical demands for adaptive queries, enabling assessment of zero-shot generalization beyond fixed labels. We report average metrics across these query types for a holistic performance overview.

During inference, given a query text, we extract the feature vector $\mathbf{e}_q$ using the BioBERT text encoder Lee et al. (2020). Gaussian language features are rasterized into a per-pixel semantic field, and a cosine similarity map $\mathbf{SC}$ is computed by comparing $\mathbf{e}_q$ with the normalized language features. A threshold $\epsilon$ (set to 0.8 in all experiments) is applied to the normalized similarity map $\mathbf{SC}^*$ to generate the segmentation mask $\mathbf{QM}$. The activated Gaussians are those whose rendering contributions (*e.g.*, with nonzero weights $w_i(x, y)$) affect pixels in $\mathbf{QM}$. Table 8 provides the detailed procedures of the language-retrieval process.

Table 8: Process of Language-driven instrument Gaussians retrieval

**Input:** language embedding map $\hat{\mathcal{F}}^l \in \mathbb{R}^{d_l \times H \times W}$ of a given view, query $q$, text encoder $\Psi_{\text{BioBERT}}$, threshold $\epsilon$

**Output:** binary mask $\mathbf{QM} \in \{0,1\}^{H \times W}$, normalized similarity map $\mathbf{SC}^* \in [0,1]^{H \times W}$, activated Gaussian primitives $\mathcal{A}$ (optional)

**Procedure**

1) $\mathbf{e}_q \leftarrow \Psi_{\text{BioBERT}}(q)$, $\hat{\mathbf{e}}_q = \mathbf{e}_q / \|\mathbf{e}_q\|_2$,   *encode query and normalize*

2) $\hat{\mathbf{f}}^l(x,y) = \hat{\mathcal{F}}^l(x,y) / \|\hat{\mathcal{F}}^l(x,y)\|_2$, $\forall (x,y)$,   *normalize per-pixel language features*

3) $\mathbf{SC}(x,y) = \langle \hat{\mathbf{f}}^l(x,y), \hat{\mathbf{e}}_q \rangle$,   *compute cosine similarity map*

4) $\mathbf{SC}^* = \dfrac{\mathbf{SC} - \min(\mathbf{SC})}{\max(\mathbf{SC}) - \min(\mathbf{SC}) + \varepsilon}$,   *optional min-max normalization*

5) $\mathbf{QM}(x,y) = \mathbb{I}\big[\mathbf{SC}^*(x,y) \geq \epsilon\big]$,   *apply threshold to obtain segmentation mask*

6) $\mathcal{A} = \{ g_i \mid \exists (x,y) \text{ s.t. } \mathbf{QM}(x,y) = 1 \text{ and } w_i(x,y) > 0 \}$,   *select activated Gaussians with nonzero rendering contribution $w_i(x,y)$*

7) return $\mathbf{QM}$, $\mathbf{SC}^*$, $\mathcal{A}$,   *outputs for 2D localization and activated primitives*

# E MORE ABLATION STUDIES

## E.1 ABLATION OF VISUAL REPRESENTATION ENHANCEMENT

We verify the effectiveness of using distilled spatial features to enhance Gaussian color representations (in Eq. 6), as shown in Fig. 13. On *StereoLung15K*, the enhancement boosts PSNR by 0.284 (from 26.44 to 26.724) and reduces RMSE by 0.487 (from 5.841 to 5.354), while mIoU remains stable (slight drop from 56.684 to 56.615). On StereoMIS, PSNR improves by 0.4 (from 28.292 to 28.692), with mIoU stable (from 59.769 to 59.767) and RMSE decreasing from 3.864 to 3.606. In all scenes, appearance reconstruction benefits from enhanced structural priors, demonstrating the module's effectiveness in reducing artifacts and improving fidelity without compromising retrieval stability.

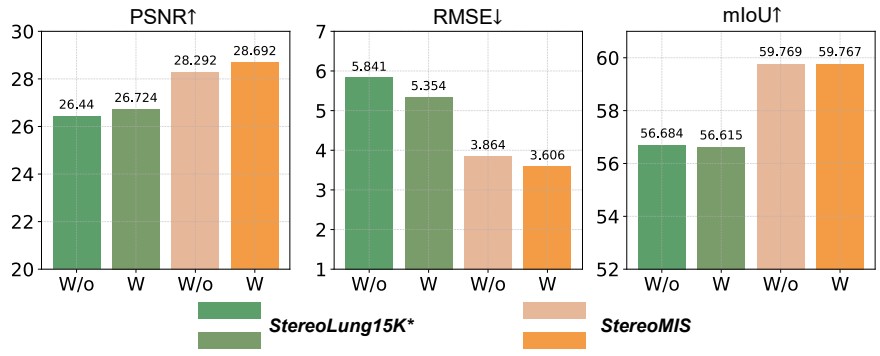

Figure 13: Ablation of visual representation enhancement

## E.2 ABLATION OF TEMPORAL MODULATION ON LANGUAGE REPRESENTATION

We investigate the efficacy of the proposed temporal modulation on language feature representations (Eq. 7), as shown in Fig. 14. Specifically, on *StereoLung15K*, temporal modulation boosts mIoU by 3.188 (from

53.427 to 56.615), mDice by 2.299 (from 61.956 to 64.255), and mRecall by 0.031 (from 0.721 to 0.752). On StereoMIS, it improves mIoU by 5.362 (from 54.405 to 59.767), mDice by 4.726 (from 60.229 to 64.955), and mRecall by 0.025 (from 0.683 to 0.708). These experimental results demonstrate that temporal modulation enhances semantic consistency in dynamic sequences, leading to consistent performance improvements across different datasets, and validating its generalizability and efficacy in capturing time-varying semantics.

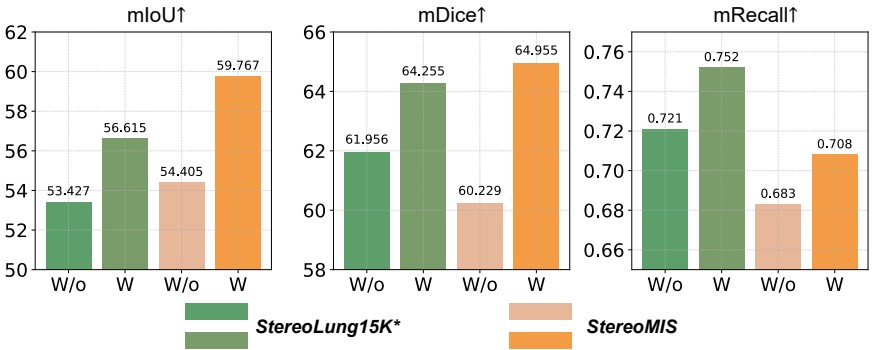

Figure 14: Ablation of temporal modulation on language representation

### E.3 IMPACT OF VARYING LLMS

To investigate the influence of different large language models (LLMs) as text encoders, we replace BioBERT with Clip-ViT-B/32 and Clip-ViT-L/14 (Radford et al., 2021), and evaluate retrieval performance on *StereoLung15K* and StereoMIS using mIoU, mDice, and mRecall metrics. As shown in Fig. 15, on *StereoLung15K*, Clip-ViT-B/32 achieves the highest mIoU (58.174) and mDice (67.421) but lower mRecall (0.705), while Clip-ViT-L/14 performs poorly (mIoU 39.181, mDice 45.945, mRecall 0.638). BioBERT achieves a balanced performance with mIoU 56.615, mDice 64.255, and the best mRecall (0.752). On StereoMIS, BioBERT outperforms with mIoU 59.767, mDice 4.955, and mRecall 0.708, compared to Clip-ViT-B/32 (mIoU 34.562, mDice 37.993, mRecall 0.439) and Clip-ViT-L/14 (mIoU 50.235, mDice 57.422, mRecall 0.621). Overall, BioBERT, fine-tuned for medical domains, consistently excels in language-driven surgical instrument retrieval, particularly in mRecall, validating its suitability for handling domain-specific queries. This makes BioBERT our default choice, as it provides the most reliable balance for surgical applications where precise recall is critical for instrument identification in complex scenes.

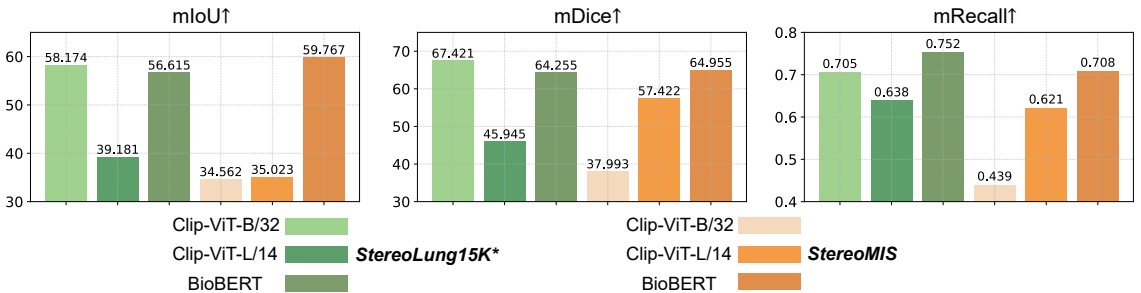

Figure 15: Impact of varying LLMs.

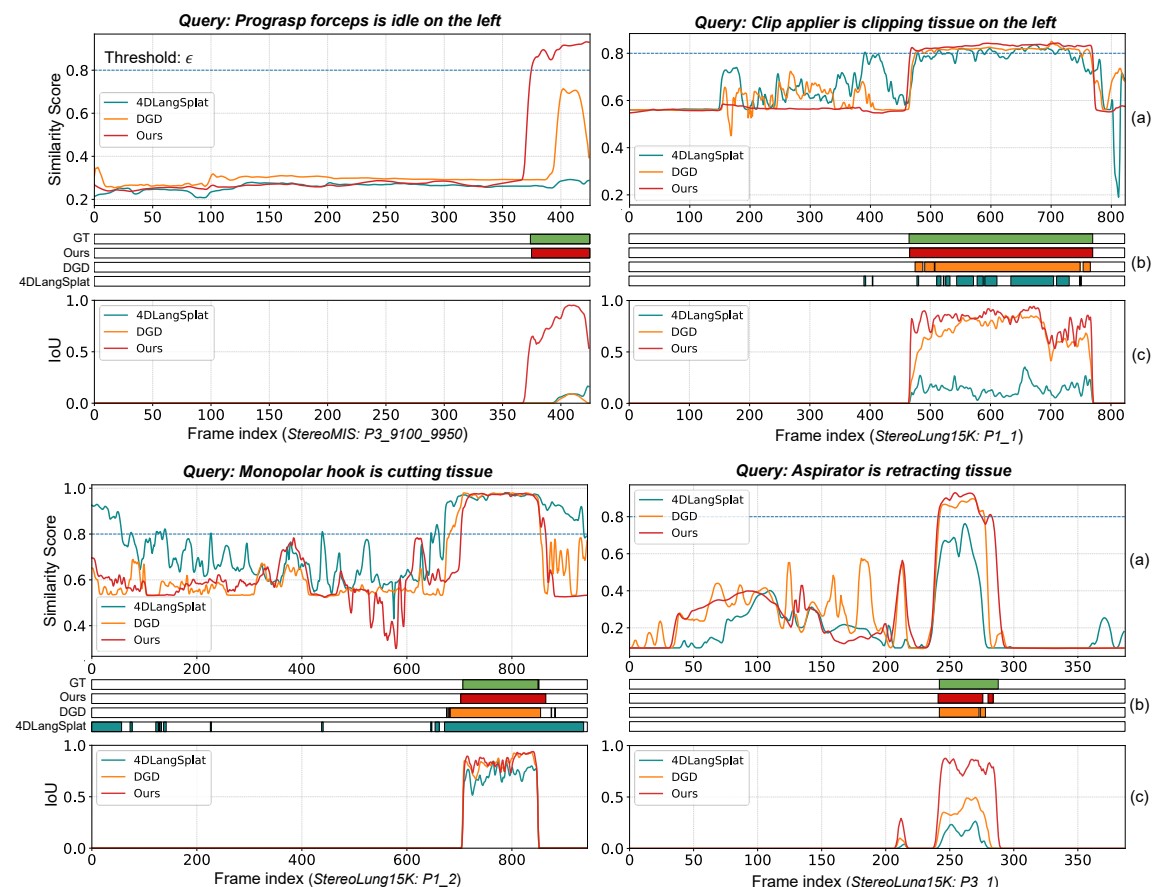

Figure 16: Quantitative comparison of scene-level surgical instrument retrieval performance on StereoMIS and *StereoLung15K* datasets. (a) Similarity score distribution across all frames for the query. (b) The horizontal bars indicate frames identified as relevant time segments for the query. Our *SurgRe4DGS* demonstrates superior performance compared to other approaches and more closely aligns with the ground truth. (c) The IoU score across all frames for the query. Our *SurgRe4DGS* shows better query performance compared to DGD (Labe et al., 2024) and 4DLangSplat (Li et al., 2025b).

## F  ADDITIONAL RESULTS

### F.1  ADDITIONAL QUANTITATIVE RESULTS

Tab. 9 and Tab. 10 give the detailed scene-wise quantitative comparison results. As we can see, our method consistently achieves better performance in photometric and geometric metrics across diverse scenes, *e.g.*, with average improvements of 1.843 in PSNR and 5.158 in SSIM over the current SOTA method Endo-Gaussian (Liu et al., 2025) for the scene $\mathcal{P}1 - 4$, 1.275 in PSNR and 4.604 in SSIM over 4DLangSplat (Li et al., 2025b) for the scene $\mathcal{P}3 - 3$, and 5.382 in RMSE over EndoRDGS (Gao et al., 2025) for the scene $\mathcal{P}3 - 6$. This superiority is attributed to our synergistic vision-language representations learning mechanism,

which reduces artifacts in deformable regions and enhances geometric accuracy, validating the framework's robustness in handling complex dynamics such as tissue motions and occlusions.

Tab. 11 and Tab. 12 present detailed scene-wise quantitative results for open-vocabulary instrument retrieval. Our proposed method demonstrates superior performance compared to other approaches, particularly with average improvements of 28.244 in mIoU and 0.327 in mRecall over the current SOTA method 4DLangSplat (Li et al., 2025b) for the scene $\mathcal{P}1 - 1$, 17.831 in mIoU and 0.213 in mRecall over DGD (Labe et al., 2024) for the scene $\mathcal{P}3 - 2$. This highlights the effectiveness of our *SurgRe4DGS* framework in free-form language-driven query tasks, as the unified vision-language representations ensure precise, consistent retrieval even under temporal variations.

Fig. 16 provides a quantitative comparison of scene-level surgical instrument retrieval across different query types. *SurgRe4DGS* consistently achieves higher retrieval accuracy, as evidenced by improved similarity score distributions (normalized to (0,1), shown in (a)), precise localization of query relevant temporal segments (shown in (b)), and more accurate instrument segmentation (shown in (c)). These results validate the effectiveness of our approach for dynamic semantic modeling and demonstrate its robustness in maintaining temporal consistency under complex surgical conditions.

## F.2 ADDITIONAL QUALITATIVE RESULTS

Fig. 17 presents additional qualitative comparison results of scene reconstruction performance on the proposed *StereoLung15K* dataset. These qualitative results highlight the superiority of our *SurgRe4DGS* in handling complex surgical conditions. For instance, in scenes with blood or smoke, our model produces sharper boundaries and more consistent textures compared to the current SOTA methods like EndoRDGS (Gao et al., 2025) and 4DLangSplat (Li et al., 2025b), which often exhibit blurring or geometric distortions. Our method outperforms the existing methods both in photometric fidelity (*e.g.*, reduced color artifacts) and geometric accuracy (*e.g.*, better preservation of 3D structures), demonstrating the efficacy and robustness of our proposed framework in dynamic, real-world surgical environments.

Fig. 18 and Fig. 19 illustrate additional comparison results of open-vocabulary instrument retrieval performance on StereoMIS and *StereoLung15K*, respectively. Across diverse queries (*e.g.*, type-only vs. type+action+position), our model accurately localizes instruments in complex surgical conditions, producing cleaner segmentation masks than competitors like DGD (Labe et al., 2024) and 4DLangSplat (Li et al., 2025b), which struggle with semantic drift in dynamic scenarios. Overall, our method outperforms the existing SOTA approaches and ensures better semantic consistency in dynamic scenarios, as evidenced by stable retrieval under temporal variations, validating the effectiveness of our unified vision-language representations for practical surgical applications.

## G BROADER APPLICATIONS, LIMITATIONS AND FUTURE WORK

Our *SurgRe4DGS* framework extends beyond surgical scene reconstruction and retrieval, with potential applications in intelligent surgical AI and robotic surgery. By fusing unified visual-language representations, it could enhance robotic systems with semantic-aware guidance for tasks like tool tracking or tissue interaction in dynamic environments. Moreover, it can facilitate advanced surgical training simulations through interactive 4D models, enabling trainees to query and visualize instrument interactions in realistic, deformable environments for improved skill development and procedural planning. In minimally invasive and resource-limited settings, such integrations could enhance efficiency and safety. Despite these advancements, our framework currently has limitations in addressing intraoperative tasks with high real-time requirements. Future work will explore advanced computing paradigms, such as cloud computing, to meet real-time demands on edge devices. Additionally, we plan to expand *StereoLung15K* by incorporating more attributes, including organ segmentation, to enable broader evaluations.

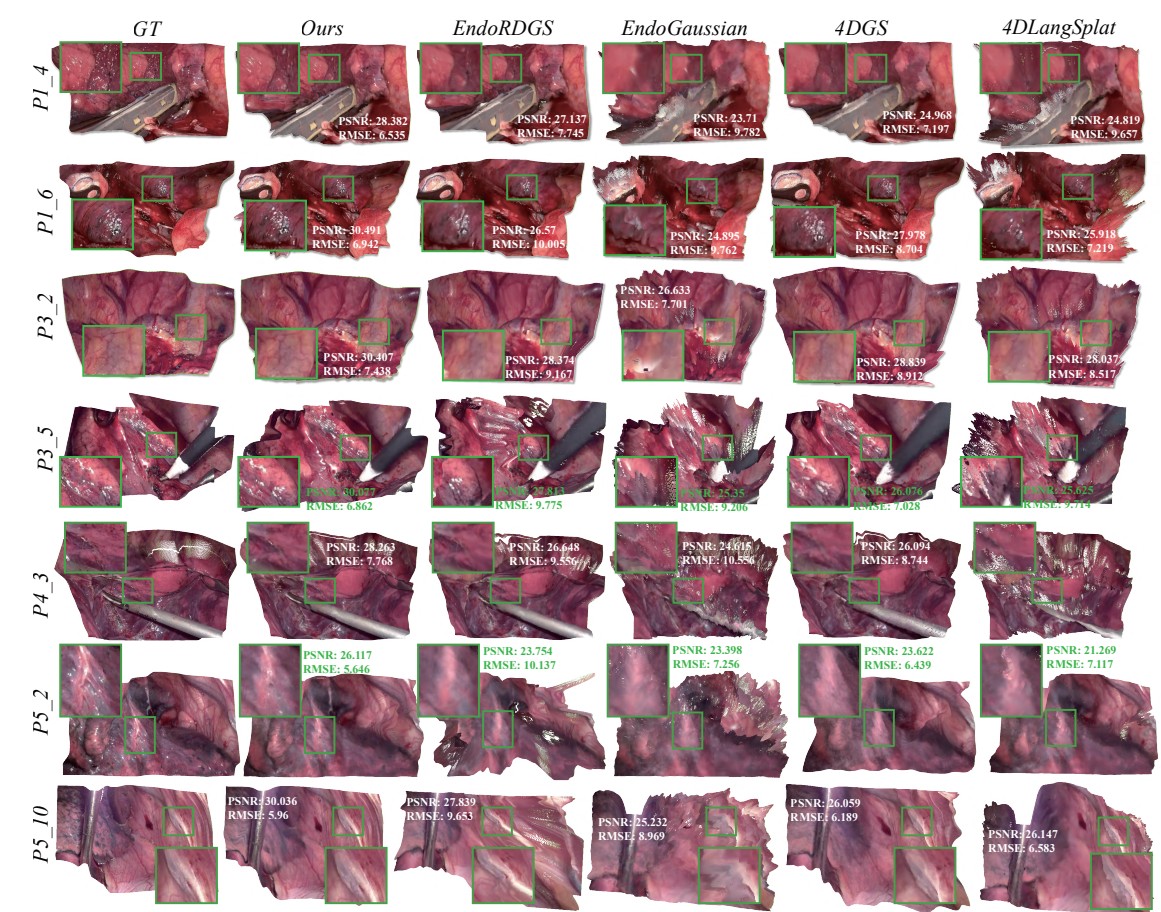

Figure 17: Additional qualitative comparisons of scene reconstruction in the *StereoLung15K* dataset.

## H  THE USE OF LARGE LANGUAGE MODEL

Apart from the specific applications detailed in the main text (*e.g.*, for generating enriched semantic descriptions via MLLM 3.2.1), large language models were solely employed as writing assistants. This involved grammar checking and minor phrasing adjustments to ensure clarity and coherence throughout the manuscript. No LLM was involved in generating technical content, experimental results, or core methodological contributions.

## I  NOTATION TABLE FOR KEY PARAMETERS AND SYMBOLS

Table 13 shows the key parameters and symbols.

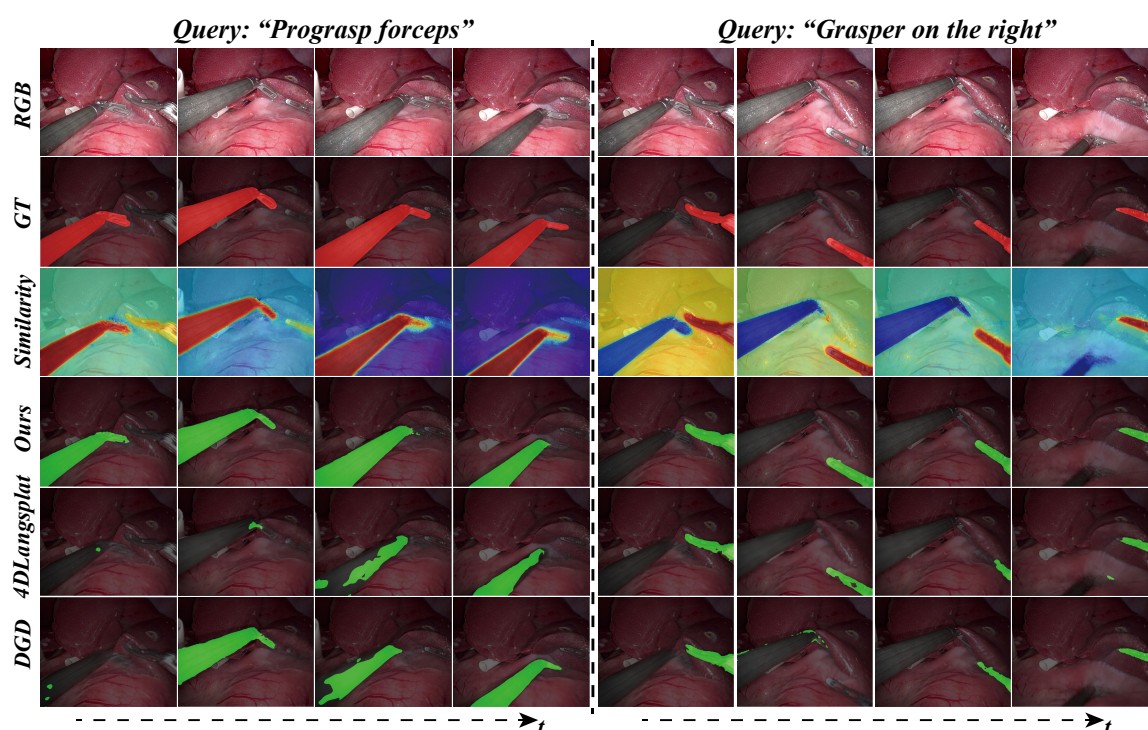

Figure 18: Additional qualitative comparisons of open-vocabulary instrument query in the StereoMIS dataset.

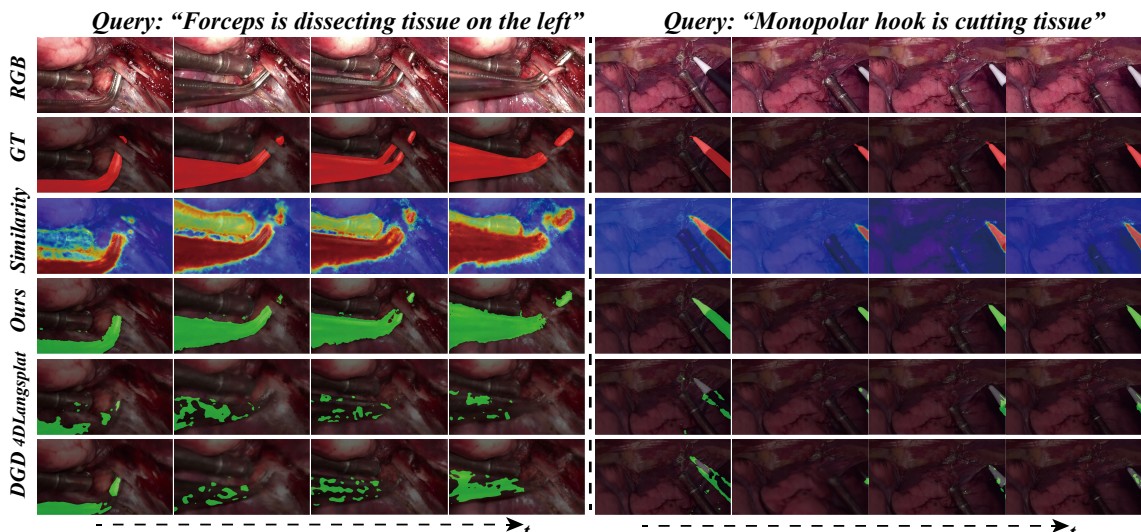

Figure 19: Additional qualitative comparisons of open-vocabulary instrument query in the *StereoLung15K* dataset.

Table 9: Scene-wise quantitative results of scene reconstruction on *StereoLung15K* dataset.

| Scene ID | Metrics | EndoGaussian Liu et al. (2025) | EndoRDGS Gao et al. (2025) | 4DGS Wu et al. (2024a) | DGD Labe et al. (2024) | 4DLangSplat Li et al. (2025b) | *SurgRe4DGS* (ours)) |
|---|---|---|---|---|---|---|---|
| $\mathcal{P}1-1$ | PSNR↑ | 22.789 | 23.852 | 23.519 | 24.048 | 23.623 | **24.387** |
| | SSIM↑ | 67.686 | 69.759 | 69.516 | **71.039** | 69.466 | 70.632 |
| | LPIPS↓ | 0.6030 | 0.5460 | 0.5910 | 0.5010 | 0.5960 | **0.4640** |
| | RMSE↓ | 4.5420 | 7.6000 | 4.5190 | - | **3.9180** | 4.2100 |
| $\mathcal{P}1-2$ | PSNR↑ | 20.124 | 20.932 | 20.619 | 21.549 | 20.684 | **21.892** |
| | SSIM↑ | 64.279 | 66.575 | 65.903 | **68.474** | 65.821 | 67.283 |
| | LPIPS↓ | 0.5740 | 0.5640 | 0.5930 | 0.5110 | 0.6050 | **0.4910** |
| | RMSE↓ | **5.7440** | 8.0400 | 6.0390 | - | 5.9230 | 5.9320 |
| $\mathcal{P}1-3$ | PSNR↑ | 19.605 | 20.706 | 20.516 | 20.596 | 20.240 | **21.321** |
| | SSIM↑ | 63.413 | 66.753 | 66.019 | **67.839** | 65.607 | 67.789 |
| | LPIPS↓ | 0.5110 | 0.5010 | 0.4930 | 0.4690 | 0.5350 | **0.4130** |
| | RMSE↓ | **3.2970** | 4.9120 | 3.5070 | - | 3.6340 | 3.4680 |
| $\mathcal{P}1-4$ | PSNR↑ | 20.599 | 21.854 | 20.891 | 21.469 | 21.201 | **22.442** |
| | SSIM↑ | 62.219 | 66.569 | 64.223 | 66.586 | 64.218 | **67.377** |
| | LPIPS↓ | 0.5330 | 0.4880 | 0.5540 | 0.4890 | 0.5410 | **0.4080** |
| | RMSE↓ | 5.9960 | 7.7210 | 6.1320 | - | 6.2710 | **5.9920** |
| $\mathcal{P}1-5$ | PSNR↑ | 20.851 | 21.863 | 21.339 | 21.561 | 21.473 | **22.612** |
| | SSIM↑ | 66.129 | 69.304 | 69.013 | 69.796 | 68.333 | **70.924** |
| | LPIPS↓ | 0.5280 | 0.4670 | 0.4790 | 0.4690 | 0.5260 | **0.3990** |
| | RMSE↓ | **6.9910** | 8.9490 | 7.3140 | - | 7.4670 | 7.0990 |
| $\mathcal{P}1-6$ | PSNR↑ | 23.308 | 24.753 | 24.209 | 24.259 | 24.119 | **25.341** |
| | SSIM↑ | 64.581 | 71.074 | 69.093 | 70.664 | 68.824 | **73.617** |
| | LPIPS↓ | 0.4870 | 0.4020 | 0.4310 | 0.3730 | 0.4510 | **0.2550** |
| | RMSE↓ | 5.4800 | 9.4340 | 5.9340 | - | **5.2780** | 5.8570 |
| $\mathcal{P}1-7$ | PSNR↑ | 20.218 | 21.234 | 20.771 | 21.107 | 20.763 | **21.669** |
| | SSIM↑ | 61.300 | 65.016 | 64.017 | **66.171** | 63.708 | 65.244 |
| | LPIPS↓ | 0.5410 | 0.4820 | 0.5140 | 0.4570 | 0.5350 | **0.4290** |
| | RMSE↓ | **5.8840** | 7.2600 | 6.9360 | - | 5.9180 | 6.2030 |
| $\mathcal{P}3-1$ | PSNR↑ | 26.763 | 28.200 | 27.293 | 27.051 | 27.398 | **29.129** |
| | SSIM↑ | 77.594 | 82.496 | 79.746 | 80.496 | 79.844 | **83.911** |
| | LPIPS↓ | 0.4270 | 0.2810 | 0.3710 | 0.3180 | 0.3680 | **0.2430** |
| | RMSE↓ | 3.8940 | 3.7480 | 3.9130 | - | 3.5490 | **3.4660** |
| $\mathcal{P}3-2$ | PSNR↑ | 24.236 | 25.459 | 24.893 | 25.985 | 25.016 | **26.253** |
| | SSIM↑ | 70.456 | 73.744 | 71.974 | **76.710** | 72.030 | 75.773 |
| | LPIPS↓ | 0.5070 | 0.4160 | 0.4910 | 0.3540 | 0.4710 | **0.3410** |
| | RMSE↓ | 6.6760 | 7.9360 | 6.9840 | - | 6.6260 | **6.1440** |
| $\mathcal{P}3-3$ | PSNR↑ | 21.297 | 22.488 | 21.991 | 22.433 | 21.867 | **23.142** |
| | SSIM↑ | 59.654 | 64.553 | 62.036 | 65.427 | 61.988 | **66.589** |
| | LPIPS↓ | 0.5450 | 0.4970 | 0.5070 | 0.4450 | 0.5220 | **0.4060** |
| | RMSE↓ | **4.8080** | 6.2500 | 5.3310 | - | 4.9540 | 5.0720 |
| $\mathcal{P}3-4$ | PSNR↑ | 21.784 | 22.923 | 22.073 | 22.281 | 22.438 | **23.698** |
| | SSIM↑ | 69.132 | 73.186 | 71.339 | 72.357 | 71.509 | **75.085** |
| | LPIPS↓ | 0.5030 | 0.4420 | 0.4910 | 0.4480 | 0.4880 | **0.3500** |
| | RMSE↓ | 7.5970 | 8.5560 | 7.9330 | - | **6.9350** | 7.5860 |
| $\mathcal{P}3-5$ | PSNR↑ | 22.807 | 24.383 | 22.439 | 23.425 | 22.999 | **25.014** |
| | SSIM↑ | 70.865 | 75.732 | 71.034 | 74.945 | 71.762 | **76.785** |
| | LPIPS↓ | 0.4780 | 0.3980 | 0.4910 | 0.4280 | 0.4840 | **0.3340** |
| | RMSE↓ | 6.0040 | 8.2580 | 6.8180 | - | 6.2820 | **5.8260** |
| $\mathcal{P}3-6$ | PSNR↑ | 27.449 | 28.890 | 28.209 | 27.634 | 28.153 | **29.590** |
| | SSIM↑ | 81.846 | 85.056 | 84.912 | 84.875 | 84.219 | **87.131** |
| | LPIPS↓ | 0.3810 | 0.3200 | 0.3080 | 0.2890 | 0.3210 | **0.1870** |
| | RMSE↓ | 4.9660 | 9.7960 | 5.4190 | - | 4.5010 | **4.4140** |

Table 10: Scene-wise quantitative results of scene reconstruction on *StereoLung15K* dataset.

| Scene ID | Metrics | EndoGaussian Liu et al. (2025) | EndoRDGS Gao et al. (2025) | 4DGS Wu et al. (2024a) | DGD Labe et al. (2024) | 4DLangSplat Li et al. (2025b) | *SurgRe4DGS* (ours)) |
|---|---|---|---|---|---|---|---|
| $\mathcal{P}4-1$ | PSNR↑ | 25.933 | 27.710 | 27.319 | 26.689 | 27.288 | **28.429** |
| | SSIM↑ | 73.018 | 79.240 | 77.337 | 77.479 | 77.169 | **80.499** |
| | LPIPS↓ | 0.4620 | 0.3300 | 0.4070 | 0.3860 | 0.4180 | **0.2920** |
| | RMSE↓ | 5.2440 | 5.0900 | 5.1130 | - | **4.5430** | 4.9030 |
| $\mathcal{P}4-2$ | PSNR↑ | 25.334 | 26.780 | 25.991 | 26.412 | 26.099 | **27.260** |
| | SSIM↑ | 71.301 | 77.470 | 74.019 | 77.762 | 74.299 | **78.765** |
| | LPIPS↓ | 0.4860 | 0.3860 | 0.4810 | 0.3680 | 0.4660 | **0.3400** |
| | RMSE↓ | 4.8900 | 4.9100 | 5.3310 | - | **4.2590** | 4.3750 |
| $\mathcal{P}4-3$ | PSNR↑ | 22.708 | 24.170 | 23.446 | 23.835 | 23.462 | **24.482** |
| | SSIM↑ | 68.124 | 73.140 | 70.801 | 71.695 | 70.742 | **73.881** |
| | LPIPS↓ | 0.5210 | 0.4610 | 0.5040 | 0.4640 | 0.5100 | **0.3900** |
| | RMSE↓ | 6.7400 | 7.9100 | 7.0390 | - | **6.6630** | 6.7840 |
| $\mathcal{P}5-1$ | PSNR↑ | 20.724 | 21.547 | 21.541 | 21.971 | 21.397 | **22.298** |
| | SSIM↑ | 66.972 | 68.723 | 69.007 | 69.581 | 68.618 | **70.138** |
| | LPIPS↓ | 0.5120 | 0.4820 | 0.4990 | 0.4880 | 0.5080 | **0.4300** |
| | RMSE↓ | 8.9380 | 10.405 | **8.3980** | - | 8.5200 | 9.3550 |
| $\mathcal{P}5-2$ | PSNR↑ | 21.315 | 22.323 | 21.891 | 22.626 | 22.076 | **23.260** |
| | SSIM↑ | 68.373 | 70.134 | 69.336 | 70.305 | 70.499 | **71.416** |
| | LPIPS↓ | 0.5220 | 0.4860 | 0.5910 | 0.5070 | 0.5610 | **0.4430** |
| | RMSE↓ | 6.2340 | 9.8020 | 7.3230 | - | 6.5150 | **6.1170** |
| $\mathcal{P}5-3$ | PSNR↑ | 20.630 | 21.735 | 21.039 | 22.079 | 21.239 | **22.450** |
| | SSIM↑ | 66.762 | 68.362 | 68.013 | 68.036 | 68.306 | **69.532** |
| | LPIPS↓ | 0.5330 | 0.4860 | 0.5540 | 0.4960 | 0.5380 | **0.4120** |
| | RMSE↓ | 6.9250 | 8.3070 | 7.4390 | - | 6.9980 | **6.7950** |
| $\mathcal{P}5-4$ | PSNR↑ | 18.714 | 18.907 | 18.079 | 19.766 | 18.211 | **19.971** |
| | SSIM↑ | 66.543 | 65.512 | 64.551 | 67.174 | 64.585 | **67.602** |
| | LPIPS↓ | 0.5860 | 0.5390 | 0.6130 | 0.5270 | 0.6460 | **0.4990** |
| | RMSE↓ | **7.4790** | 10.185 | 9.6640 | - | 9.6710 | 8.0540 |
| $\mathcal{P}5-5$ | PSNR↑ | 19.827 | 20.931 | 20.039 | 20.752 | 20.266 | **21.512** |
| | SSIM↑ | 59.514 | 63.150 | 61.143 | 64.491 | 61.441 | **64.893** |
| | LPIPS↓ | 0.5260 | 0.4770 | 0.5540 | 0.5050 | 0.5420 | **0.4120** |
| | RMSE↓ | 6.5890 | 8.5590 | 7.3590 | - | 6.9250 | **6.3520** |
| $\mathcal{P}5-6$ | PSNR↑ | 19.673 | 20.700 | 19.504 | 20.438 | 19.965 | **21.138** |
| | SSIM↑ | 63.293 | 65.727 | 64.317 | 66.873 | 63.915 | **66.936** |
| | LPIPS↓ | 0.5540 | 0.5070 | 0.5380 | 0.5220 | 0.5460 | **0.4190** |
| | RMSE↓ | 6.8750 | 9.5530 | **6.0310** | - | 6.6680 | 6.8390 |
| $\mathcal{P}5-7$ | PSNR↑ | 20.223 | 21.117 | 20.807 | 21.327 | 20.778 | **21.683** |
| | SSIM↑ | 60.335 | 62.561 | 62.318 | 62.985 | 62.074 | **63.685** |
| | LPIPS↓ | 0.5500 | 0.5230 | 0.5170 | 0.5200 | 0.5520 | **0.4560** |
| | RMSE↓ | 7.7460 | 9.1880 | **7.4910** | - | 7.6530 | 7.4970 |
| $\mathcal{P}5-8$ | PSNR↑ | 22.173 | 23.009 | 22.591 | 22.592 | 22.577 | **23.697** |
| | SSIM↑ | 68.672 | 72.166 | 70.584 | 72.562 | 70.507 | **74.020** |
| | LPIPS↓ | 0.5020 | 0.4860 | 0.5070 | 0.4550 | 0.5150 | **0.4110** |
| | RMSE↓ | **7.6340** | 9.9350 | 8.9910 | - | 8.4160 | 7.8830 |
| $\mathcal{P}5-9$ | PSNR↑ | 21.613 | 22.318 | 22.193 | 22.344 | 22.176 | **23.109** |
| | SSIM↑ | 71.472 | 72.914 | **73.917** | 72.361 | 73.652 | 73.891 |
| | LPIPS↓ | 0.5430 | 0.5020 | 0.5730 | 0.4970 | 0.5830 | **0.4670** |
| | RMSE↓ | 6.6990 | 9.8580 | 6.4920 | - | 6.2050 | **6.2010** |
| $\mathcal{P}5-10$ | PSNR↑ | 22.943 | 24.356 | 23.913 | 24.139 | 23.691 | **25.150** |
| | SSIM↑ | 70.786 | 74.425 | 72.818 | 75.201 | 72.749 | **75.571** |
| | LPIPS↓ | 0.4720 | 0.4450 | 0.4710 | 0.4570 | 0.4810 | **0.3550** |
| | RMSE↓ | **7.3480** | 10.059 | 7.9380 | - | 7.5640 | 7.4510 |

Table 11: Scene-wise quantitative results of language-driven tool retrieval on *StereoLung15K* dataset.

| Scene ID | Metrics | SurgTPGS Huang et al. (2025) | DGD Labe et al. (2024) | 4DLangSplat Li et al. (2025b) | *SurgRe4DGS* (ours)) |
|---|---|---|---|---|---|
| $\mathcal{P}1-1$ | mIoU↑ | 38.730 | 40.954 | 19.288 | **47.532** |
| | mDice↑ | 49.990 | 51.489 | 29.260 | **57.118** |
| | mRecall↑ | 0.4380 | 0.5390 | 0.2790 | **0.6060** |
| $\mathcal{P}1-2$ | mIoU↑ | 43.127 | 58.664 | 61.689 | **66.293** |
| | mDice↑ | 53.319 | 68.939 | 66.115 | **74.918** |
| | mRecall↑ | 0.5720 | 0.6170 | 0.6890 | **0.7290** |
| $\mathcal{P}1-3$ | mIoU↑ | 40.704 | 63.679 | 68.924 | **72.401** |
| | mDice↑ | 50.189 | 73.066 | 76.118 | **80.630** |
| | mRecall↑ | 0.5950 | 0.7230 | 0.7920 | **0.8360** |
| $\mathcal{P}1-4$ | mIoU↑ | 53.776 | 56.936 | 42.521 | **68.293** |
| | mDice↑ | 60.432 | 64.575 | 49.392 | **75.873** |
| | mRecall↑ | 0.6320 | 0.5850 | 0.4310 | **0.7310** |
| $\mathcal{P}1-5$ | mIoU↑ | 30.108 | 50.113 | 56.446 | **61.068** |
| | mDice↑ | 41.092 | 59.629 | 63.169 | **69.756** |
| | mRecall↑ | 0.3500 | 0.5250 | 0.5790 | **0.6910** |
| $\mathcal{P}1-6$ | mIoU↑ | 45.408 | 78.630 | 67.727 | **82.871** |
| | mDice↑ | 51.588 | 84.579 | 75.649 | **88.194** |
| | mRecall↑ | **0.8750** | 0.8070 | 0.7110 | 0.8660 |
| $\mathcal{P}1-7$ | mIoU↑ | 50.528 | 40.077 | 57.913 | **60.151** |
| | mDice↑ | 59.303 | 49.318 | 65.039 | **69.593** |
| | mRecall↑ | 0.5380 | 0.4220 | 0.6370 | **0.6530** |
| $\mathcal{P}3-1$ | mIoU↑ | 65.906 | 59.051 | 66.192 | **70.626** |
| | mDice↑ | 76.407 | 71.270 | 78.332 | **81.327** |
| | mRecall↑ | 0.7020 | 0.6150 | 0.6870 | **0.7340** |
| $\mathcal{P}3-2$ | mIoU↑ | 46.174 | 51.364 | 45.981 | **69.195** |
| | mDice↑ | 51.452 | 58.199 | 54.582 | **75.914** |
| | mRecall↑ | 0.7340 | 0.5270 | 0.5260 | **0.7400** |
| $\mathcal{P}3-3$ | mIoU↑ | **47.211** | 41.039 | 29.230 | 42.215 |
| | mDice↑ | **56.627** | 47.592 | 37.110 | 49.136 |
| | mRecall↑ | 0.5580 | 0.7050 | 0.6290 | **0.7760** |
| $\mathcal{P}3-4$ | mIoU↑ | **54.120** | 34.408 | 45.346 | 49.341 |
| | mDice↑ | **62.285** | 40.789 | 51.153 | 56.806 |
| | mRecall↑ | 0.5770 | 0.3640 | 0.6290 | **0.6710** |
| $\mathcal{P}3-5$ | mIoU↑ | 63.103 | 73.600 | 68.374 | **74.336** |
| | mDice↑ | 68.477 | 80.348 | 78.328 | **81.995** |
| | mRecall↑ | **0.8780** | 0.7660 | 0.7940 | 0.8350 |
| $\mathcal{P}3-6$ | mIoU↑ | 46.614 | 53.126 | 59.324 | **61.423** |
| | mDice↑ | 52.558 | 61.244 | 64.369 | **68.105** |
| | mRecall↑ | 0.5880 | 0.6230 | 0.6920 | **0.7240** |

Table 12: Scene-wise quantitative results of language-driven tool retrieval on *StereoLung15K* dataset.

| Scene ID | Metrics | SurgTPGS Huang et al. (2025) | DGD Labe et al. (2024) | 4DLangSplat Li et al. (2025b) | *SurgRe4DGS* (ours)) |
|---|---|---|---|---|---|
| $\mathcal{P}4-1$ | mIoU↑ | 49.199 | 60.138 | 40.711 | **69.079** |
| | mDice↑ | 54.323 | 68.068 | 50.022 | **76.189** |
| | mRecall↑ | 0.5900 | 0.6360 | 0.4220 | **0.7520** |
| $\mathcal{P}4-2$ | mIoU↑ | 24.620 | 31.338 | 27.317 | **33.011** |
| | mDice↑ | 30.552 | 39.074 | 35.522 | **40.186** |
| | mRecall↑ | 0.5330 | 0.6640 | 0.6250 | **0.7230** |
| $\mathcal{P}4-3$ | mIoU↑ | 41.812 | 58.358 | 31.439 | **67.755** |
| | mDice↑ | 52.288 | 68.016 | 40.843 | **76.391** |
| | mRecall↑ | 0.5870 | 0.6540 | 0.3510 | **0.7820** |
| $\mathcal{P}5-1$ | mIoU↑ | 32.919 | 32.907 | 37.231 | **41.638** |
| | mDice↑ | 40.591 | 41.651 | 42.179 | **49.731** |
| | mRecall↑ | 0.5060 | 0.5000 | 0.5810 | **0.6270** |
| $\mathcal{P}5-2$ | mIoU↑ | 36.879 | 27.499 | 38.625 | **40.452** |
| | mDice↑ | 47.601 | 36.227 | 43.192 | **49.951** |
| | mRecall↑ | 0.5180 | 0.3450 | 0.4910 | **0.5480** |
| $\mathcal{P}5-3$ | mIoU↑ | **41.792** | 38.840 | 33.189 | 41.658 |
| | mDice↑ | **50.098** | 45.651 | 40.449 | 48.296 |
| | mRecall↑ | 0.5790 | 0.7430 | **0.7510** | 0.7410 |
| $\mathcal{P}5-4$ | mIoU↑ | 14.862 | **38.220** | 22.123 | 26.242 |
| | mDice↑ | 22.127 | **46.888** | 29.771 | 33.984 |
| | mRecall↑ | 0.2860 | **0.5420** | 0.3160 | 0.4950 |
| $\mathcal{P}5-5$ | mIoU↑ | 49.549 | 60.052 | 34.375 | **61.492** |
| | mDice↑ | 60.689 | 68.661 | 45.068 | **70.581** |
| | mRecall↑ | 0.7160 | 0.7990 | 0.5170 | **0.8470** |
| $\mathcal{P}5-6$ | mIoU↑ | **65.858** | 49.465 | 35.178 | 54.794 |
| | mDice↑ | **74.936** | 59.266 | 47.094 | 63.239 |
| | mRecall↑ | 0.8050 | 0.7740 | 0.6070 | **0.8420** |
| $\mathcal{P}5-7$ | mIoU↑ | 41.916 | 47.765 | 51.177 | **60.637** |
| | mDice↑ | 51.659 | 59.943 | 55.625 | **71.598** |
| | mRecall↑ | **0.7170** | 0.5010 | 0.593 | 0.6630 |
| $\mathcal{P}5-8$ | mIoU↑ | 32.114 | 44.242 | 30.936 | **46.375** |
| | mDice↑ | 37.077 | 51.137 | 37.831 | **52.179** |
| | mRecall↑ | 0.3550 | 0.5150 | 0.4050 | **0.5700** |
| $\mathcal{P}5-9$ | mIoU↑ | 45.779 | 43.240 | 29.571 | **54.224** |
| | mDice↑ | 56.045 | 52.677 | 39.811 | **63.072** |
| | mRecall↑ | 0.5570 | 0.6690 | 0.5580 | **0.8070** |
| $\mathcal{P}5-10$ | mIoU↑ | 32.714 | 36.897 | 53.514 | **55.583** |
| | mDice↑ | 38.156 | 44.186 | 59.671 | **65.087** |
| | mRecall↑ | **0.6590** | 0.3940 | 0.6360 | 0.6130 |

Table 13: Notation Table for Key Parameters and Symbols

| Symbol | Definition |
|---|---|
| $g^i$ | A single 3D Gaussian primitive (from the set $\mathcal{G}$) |
| $\mu^i, \Sigma^i, o^i, c^i$ | The position, covariance, opacity, and base color features of $g^i$ |
| $z_i = [f_i^v, f_i^l]$ | Our **novel unified latent vector** attached to each Gaussian $g^i$ |
| $f_i^v \in \mathbb{R}^{d_v}$ | The **visual feature** subspace of $z_i$ (for reconstruction) |
| $f_i^l \in \mathbb{R}^{d_l}$ | The **language feature** subspace of $z_i$ (for retrieval) |
| $\Psi_{MLLM}$ | The Multimodal Large Language Model used for semantic enrichment (Sec. 3.2.1) |
| $\mathcal{F}^v$ | The 2D target visual feature map (distilled from SAM) |
| $\mathcal{F}^l$ | The 2D target language feature map (from MLLM-enriched expressions) |
| $\hat{\mathcal{F}}^v, \hat{\mathcal{F}}^l$ | The **rendered** 2D visual and language feature maps (from Eq. 4) |
| $\Gamma_v, \Gamma_l$ | Our **Temporal Modulation** MLPs for the visual and language branches |
| $c_i^*$ | The final, time-modulated color feature used for rendering (Eq. 6) |
| $\mathcal{L}_{color}, \mathcal{L}_{depth}$ | Standard photometric and depth reconstruction losses (Eq. 9) |
| $\mathcal{L}_v$ | The visual feature alignment loss (L1 loss between $\hat{\mathcal{F}}^v$ and $\mathcal{F}^v$) |
| $\mathcal{L}_l$ | Our **novel Hybrid Semantics Injection Loss** (Eq. 9) |
| $\mathcal{L}_{render}^l$ | Coarse semantic alignment loss (L2 loss on the rendered map $\hat{\mathcal{F}}^l$) |
| $\mathcal{L}_{align}^l$ | Our **novel weighted top-K alignment loss** (cosine similarity) |
| $\mathcal{L}_{cons}^l$ | Our **novel cross-instance contrastive loss** (Eq. 8) |
| $K$ | The number of top contributing Gaussians used in $\mathcal{L}_{align}^l$ |

