# OpenReview forum: "SurgRe4DGS: Open-Vocabulary Vision-Language Gaussian Splatting for Retrievable 4D Surgical Scene Reconstruction"
_ICLR.cc/2026/Conference — ICLR 2026 Conference Desk Rejected Submission_

### Official Review · Reviewer_ezT5 · 2025-10-23

**Soundness:** 3
**Presentation:** 4
**Contribution:** 2
**Rating:** 4
**Confidence:** 4

**Summary:**

The manuscripts presents a method for reconstructing dynamic surgical scenes for open-vocabulary retrieval tasks using a synergistic vision-language Gaussian representation, called SurgRe4DGS. This is potentially important for robotic-assisted surgery, where understanding the tool-to-tissue interactions is crucial for potental applications such as automation, training, and guidance. The method uses spatial color blending with temporal modulation to enhance color represenations, while language features depend on instrument-wise segmentation and interpretation by a multi-modal large language model (MLLM), with text encoding of the resultant description being used as the language features. This enables query-time retrieval of specific surgical tools in the 4D scene as well as novel view synthesis. The method is evaluated on StereoMIS, EndoNerf, and a novel bronchoscopy dataset, StereoLung15K, with results showing improved performance over prior methods in terms of novel view quality (PSNR, SSIM, LPIPS, etc) and open vocabulary instrument retrieval.

**Strengths:**

\- The paper is well-written and detailed, with a detailed explanation of Gaussian splatting prerequisites, the proposed method, and the experimental setup.
\- The novel dataset, StereoLung15K, is a valuable contribution to the field, providing a new benchmark for evaluating 4D reconstruction and retrieval methods in bronchoscopy scenarios.
\- The method outperforms prior methods for novel view synthesis on multiple datasets, including a new bronchoscopy dataset introduced in the paper, achieving a PSNR gain of 2 \- 3 % over the strongest baseline, for example.
\- For instrument retrieval, the method performs significantly better than related methods in terms of mIoU, mDice, and mRecall, especially on StereoLung15kK and StereoMIS. On StereoLung15K, for example, the mIoU is improved from 49.9 to 57.9.

**Weaknesses:**

- The "open-vocabulary" retrieval is limited to surgical tools only, considerably narrowing the scope and potential applications of the method. Any surgical robot can automatically determine the tools being used in the procedure based the electronic tags associated with the instruments, making it a simple matter to retrieve tool segmentations in 2D images using the frozen tool models mentioned in the paper and associate them with 3D locations using available 4D reconstruction methods. The real value of open-vocabulary retrieval would be the ability to retrieve instruments and tissue types both.
- Relatedly, the use of 4DGS in the context of RAS is poorly motivated. The abstract claims that 4D reconstruction of surgical workflows is **crucial** for robot-assisted surgery, but this is clearly not the case, because current surgical robots operate without any 4D reconstruction capabilities. Intra-operative augmented reality applications, as mentioned in the introduction, may benefit from the open-vocabulary retrieval and novel view synthesis capabilities, but the connection between "capturing tissue deformation, instrument motion,..." and any practical application is not made clear, especially given the limits of the method to surgical tools. What specific augmented reality application would benefit from the ability to retrieve surgical tools in 4D space? The surgeon can clearly see where the tools are without any AR assistance; rather, the location of anatomical features would be more useful.
- The approach has limited novelty compared to prior work, particularly 4D LangSplat [1], which also uses MLLMs to generate language features for Gaussian splatting-based 4D reconstruction.

1. Li, Wanhua, et al. "4D LangSplat: 4D Language Gaussian Splatting via Multimodal Large Language Models." arXiv, 13 Mar. 2025, doi:10.48550/arXiv.2503.10437.

**Questions:**

1\. How well does the method generalize to surgical tools not seen during training? Can the authors provide quantitative results on this aspect?
2\. Given the limited number of surgical tools typically used in RAS procedures, what is the practical advantage of open-vocabulary retrieval of surgical tools compared to using known tool models to identify and retrieve them in 4D space?
3\. Can the authors clarify the specific clinical applications that would benefit from the proposed method, especially considering its limitation to surgical tools only?
4\. Can the authors ensure that the dataset (StereoLung15K) will be made public?

---

> ### Author Response · Authors · 2025-11-18
> **Response to Reviewer ezT5 (1/3)**
>
> Dear reviewer ezT5, thank you for your detailed feedback and for your high praise of our paper's execution, rating our Presentation as "*Excellent*" and acknowledging our "*well-written and detailed*" explanation. We are also grateful that the reviewer recognized our strong quantitative results, noting that our method "*outperforms prior methods*" and is "*significantly better*" on retrieval tasks.
>
> ---
> **Q1:  Clarification on scope and motivation of tool-only retrieval.**
>
> **A1:** We thank the reviewer for this critical question on clinical motivation. First, we would like to clarify that our work is a foundational exploration in a new direction. The reviewer noted our StereoLung15K dataset is a "valuable contribution". We would add that it is the first large-scale benchmark to provide the dense, multimodal annotations required for this joint 4D reconstruction and language-retrieval task. Our paper represents the first systematic exploration of this problem. Second, the reviewer's critique focuses on the narrow application of robotic surgery (RAS). We would like to clarify that RAS is only one potential, future application of our work. Our framework targets the much broader ecosystem of Surgical AI and intelligent auxiliary systems. This ecosystem applies to all minimally invasive surgeries, including the vast majority of non-robotic, manual procedures where "electronic tags" do not exist. In this broader Surgical AI context, the clinical value lies in postoperative analysis, video indexing, and surgical training. Our framework enables, for the first time, a semantic 4D search of the surgical workflow. This is a foundational capability for surgical analytics, skill assessment, and training. While retrieving tissue is the ultimate goal, as the reviewer notes, understanding the tool-tissue interaction is the necessary first step. That is the novel contribution we have enabled.
>
> ---
> **Q2: Clarification on motivation and application for 4DGS in surgical.**
>
> **A2:** We thank the reviewer for this insightful critique, which allows us to clarify the core motivation of our work and intended scope. We agree with the reviewer's premise that current commercial robotic systems do not yet utilize 4D reconstruction. Our claim that it is "crucial" refers to its foundational role in enabling the future of robotic systems and context-aware Surgical AI, which our work aims to unlock. The reviewer asks for the practical connection. The primary contribution of our work is the 4D reconstruction itself: creating a high-fidelity, temporally-aware model of the surgical scenario. The instrument retrieval is the novel interface we developed to make this complex 4D data searchable and understandable for the first time.
> This high-fidelity 4D reconstruction has direct, practical applications. For instance, a key challenge in surgical AR is registering a static, pre-operative 3D anatomical model (from a CT scan) to the live, deforming anatomy within the surgical site. Our framework's ability to accurately "capture tissue deformation" provides the real-time, deformable registration necessary for this task. This enables the AR overlay of hidden anatomy (like vessels). Furthermore, this 4D reconstruction serves as a digital twin for postoperative analysis and training. Our language retrieval is the tool that unlocks this value. A surgeon cannot manually scrub through a lengthy 4D file, but they can now use our method to ask semantic questions about the motion and interactions within that 4D model. Therefore, we view the 4D reconstruction as the crucial foundation, and our language retrieval as the novel tool that makes this foundation usable and accessible.

---

> ### Author Response · Authors · 2025-11-18
> **Response to Reviewer ezT5 (2/3)**
>
> ---
> **Q3: Clarification on novelty compared to 4DLangSplat.**
>
> **A3:** We thank the reviewer for this question. We respectfully disagree that our novelty is limited, and the proof is in both our methodology and our results.
> + **Different methodology:** The reviewer is correct that both methods use MLLMs, but how they are used is fundamentally different. 4DLangSplat uses an MLLM to generate general video captions for objects. In contrast, our method uses an MLLM in a much more fine-grained, structured way: to enrich per-instance semantic attributes.
> + **Novel semantics integration strategy:** 4DLangSplat relies on standard volume rendering to align features, which inevitably leads to semantic ambiguity and boundary blurring in complex surgical scenes. In contrast, our method introduces a "Hybrid Semantics Injection Scheme" designed for surgical precision. This scheme integrates (1) weighted top-K alignment ($\mathcal{L}\_{align}^{l}$) to robustly handle noisy depth and (2) cross-instance contrastive loss ($\mathcal{L}\_{cons}^{l}$) to resolve ambiguity, tackling problems 4DLangSplat does not address.
> + **Better performance and Capability:** The clearest proof of our novelty is not just the ~14 mIoU absolute gain (57.907 vs 43.914 on StereoLung15K), but the specific capabilities enabled by our method. 4DLangSplat struggles to distinguish between multiple identical instruments or handle complex spatial queries due to the lack of instance-level perception. As shown in Figure 6 and Figure 19, this leads to false positives or ambiguous localization in 4DLangSplat results. In contrast, our method precisely isolates the target instrument in these challenging scenarios, demonstrating robustness that 4DLangSplat cannot achieve.
>
> ---
> **Q4: On generalization to unseen tools.**
>
> **A4:** We thank the reviewer for this question. We would like to clarify that our method, in line with 4DGS and related reconstruction methods, is a per-scene optimization framework. It is designed to be trained offline on the contents of a specific video to create a high-fidelity 4D model of that particular scene. While zero-shot generalizability to unseen instruments is an interesting future research direction, it is not the focus of this work. We will consider this as a valuable direction for our future work.
>
> ---
> **Q5: On the practical advantage of open-vocabulary vs. known tool models.**
>
> **A5:** We thank the reviewer for this question. It is correct that if the goal were merely to identify the class of a known tool (e.g., "grasper") from a limited set, a standard closed-set segmentation model would be sufficient. However, our primary goal is to understand the surgical workflow within our high-fidelity 4D reconstruction. The practical advantage of our open-vocabulary system is that it moves beyond simple identification to enable context-aware semantic retrieval. A closed-set model is rigid; it cannot differentiate between "an idle grasper" and "a grasper cutting tissue." Our method, by training on MLLM-enriched, multi-attribute descriptions, learns a rich semantic embedding space. This allows our single framework to serve as a flexible and natural interface for our 4D reconstruction, enabling users to perform complex, contextual queries. This ability to perform semantic 4D search is the key practical advantage. It is the tool that makes our 4D reconstruction (the core asset) truly usable for the postoperative analysis, video indexing, and training applications we discussed in A1 and A2.

---

> ### Author Response · Authors · 2025-11-18
> **Response to Reviewer ezT5 (3/3)**
>
> ---
> **Q6: Clarification on specific clinical applications and the "limitation to tools only".**
>
> **A6:** We thank the reviewer for this question. The "limitation to surgical tools only" is not a methodological constraint, but rather a reflection of the currently available training data. Our framework is a general 4D retrievable 4D surgical reconstruction system designed to learn a mapping between any visual features and their corresponding semantic embeddings. While the ideal scenario, as the reviewer suggests, would be to retrieve complex "tool-action-tissue" events (e.g., "grasper retracting lung lobe"), the problem is that such a dataset (with high-frame-rate tool-action-tissue annotations) does not exist in the field. Our StereoLung15K dataset, which the reviewer noted as a "valuable contribution", is the first benchmark to provide the necessary dense, multi-modal annotations for this task. We therefore used tool-action-position as the first, concrete exemplar to validate our general framework.
> Thus, the specific clinical applications are not limited to tools. The primary application is the combination of the 4D reconstruction itself (the core asset) and the general semantic 4D search (the interface). As discussed in A1 and A2, the reconstruction serves as a digital twin for postoperative analysis and AR registration, while the semantic search is the tool that makes this complex 4D data usable for video indexing and training. Our framework is already designed to handle these more complex queries, and we see the future creation of tool-action-tissue datasets as the clear next step to unlock its full potential.
>
> ---
> **Q7: On dataset release.**
>
> **A7:** We thank the reviewer for asking for this clarification. Yes, absolutely.
> We unambiguously commit to the full public release of the entire StereoLung15K dataset upon acceptance. This commitment is already stated in our Abstract and our Reproducibility Statement. We have already obtained full ethics approval (IRB) from the collaborating hospitals for this public release. To further ensure reproducibility and allow for immediate verification, we have also provided an anonymous GitHub repository (please refer to A1 of reviewer ADhH). This repository contains the source code of our work, two complete sample data cases, and their corresponding pre-trained GS models.
>
> ---
> We thank the reviewer again for challenging us to clarify our motivation. We hope our response regarding the foundational role of 4D reconstruction and the broader scope of Surgical AI resolves the concerns regarding our contribution. We believe our work serves as a crucial stepping stone for the field and hope this warrants a positive re-evaluation.

---

### Official Review · Reviewer_ADhH · 2025-10-26

**Soundness:** 3
**Presentation:** 3
**Contribution:** 3
**Rating:** 4
**Confidence:** 5

**Summary:**

The paper proposes SurgRe4DGS, a vision-language framework for retrievable 4D surgical scene reconstruction using Gaussian Splatting. It introduces a dual-branch architecture combining visual priors and language semantics via MLLM-guided embeddings to enable open-vocabulary instrument retrieval alongside dynamic reconstruction. The authors also present StereoLung15K, a new high-frame-rate multimodal dataset with RGB-D-text annotations. Experiments show strong gains over recent 4DGS and retrieval baselines. I firmly urge the authors to release the dataset publicly to ensure fair evaluation and to benefit the community. I think this paper presents a strong idea, a solid framework, but incomplete execution. I still maintain a positive attitude toward this paper, provided the authors can demonstrate strong reproducibility, as this work has the potential to contribute significantly to this field

**Strengths:**

This paper introduces unified vision-language Gaussian splatting for 4D surgery. and it outperforms baselines across 4D reconstruction and retrieval tasks. Another contribution is presenting StereoLung15K: a multimodal, densely annotated surgical dataset. This paper also provides thorough ablation on architectural components and semantic injection, which is appreciated.

**Weaknesses:**

Though the authors present substantial experiments, several issues raise concern:
1. No video results or temporal visualizations undermine the 4D claims, and no code or dataset is released, contradicting reproducibility expectations. I strongly encourage any paper in this field to provide reproducible code and substantial video comparisons to ensure a fair evaluation.
2. Experiments use only a few sequences per dataset; robustness across diverse surgical conditions, unseen instruments, or more complex anatomy remains unproven.
3. No runtime or inference latency is reported, and the pipeline involves high computational overhead: it depends on a large vision model (SAM), a multi-modal language model (MLLM), Gaussian rendering(this is fast though), and contrastive training. As presented, this appears to be an offline process requiring significant compute. It is unclear whether the system is usable for real-time or even near-time surgical applications.
4.The method relies on pre-calibrated camera poses, yet makes no reference to surgical SLAM systems (e.g., EndoSLAM, EndoGSLAM), which directly address pose estimation in deformable surgical scenes. These are foundational to the assumptions of this paper and must be cited.
5.Although the method claims open-vocabulary capability, the model is trained only on structured, attribute-based annotations from the StereoLung15K dataset (instrument type, action, position). It remains unclear how well the system generalizes to unseen or natural-language queries, limiting the demonstrated scope of retrieval.
6. The evaluation focuses on reconstruction quality and retrieval accuracy but omits practical metrics like retrieval latency or long-term temporal stability. These are critical for assessing real-world surgical utility.

**Questions:**

1. Why are there no videos or visualizations of reconstructed sequences?
1. Will the authors release code, StereoLung15K, and trained models?
1. How does the system generalize to unseen surgical instruments or phrasing?

---

> ### Author Response · Authors · 2025-11-18
> **Response to Reviewer ADhH (1/3)**
>
> Dear reviewer ADhH, thank you for your thoughtful review and constructive suggestions. In the following, we respond to your questions in detail:
>
> ---
> **Q1: On reproducibility (videos, code, and data).**
>
> **A1:** We thank the reviewer for this critical feedback. We agree that reproducibility is paramount for a fair evaluation and for the benefit of the community. To fully address the reviewer's concerns, we have provided an anonymous repository [Anonymous Repository Link](https://anonymous.4open.science/r/ICLR26-69B0-zxcvbnm934) which contains all necessary materials for a thorough evaluation. This repository includes:
> + **Video demonstrations:** We have included two comprehensive video demos to support our 4D claims: (1) A comparison of our 4D scene reconstruction (showing novel view fly-throughs and temporal stability). (2) A demonstration of our live instrument retrieval, showing the mask updating in real-time to different text queries.
> + **Full codebase and verifiable data cases:** To ensure our work is fully reproducible, we have uploaded: (a) The source code for our work. (2) Two complete sample data cases from our dataset. (3) The corresponding pre-trained GS models for these samples, allowing for immediate verification of our results.
> + **Full dataset release:** Regarding the full StereoLung15K dataset, we unambiguously commit to its full public release upon acceptance. We have already obtained full ethics approval (IRB) from the collaborating hospitals. However, to prevent potential data leakage via unsecured public links during the review process, a controlled post-acceptance release is the standard and secure approach.

---

> ### Author Response · Authors · 2025-11-18
> **Response to Reviewer ADhH (2/3)**
>
> ---
> **Q2: On generalizability, robustness, and number of sequences.**
>
> **A2:** We thank the reviewer for this comment and would like to clarify our experimental paradigm, as this addresses the core of the concern. First, our method, in line with the standard paradigm of 4D Gaussian Splatting (e.g., 4DGS ), is an off-line optimization framework. Our primary goal is to enhance the reconstruction quality and semantic understanding for each given scene, which is the established goal of this research field. While zero-shot generalizability to unseen scenes is an interesting future research direction, it is not the focus of this work. Second, to address the reviewer's valid concern about the robustness of our methodology, we did not test on "only a few sequences." Instead, we applied our single framework to all available sequences across three highly diverse datasets, totaling **35 distinct surgical scenes**. These datasets cover different surgical types and organs (thoracic lung surgery, robotic prostatectomy, and abdominal/pelvic), and our method achieved SOTA results consistently across all of them. Furthermore, our evaluation did include the challenging scenarios the reviewer mentioned: (1) **Diverse surgical conditions (Noise):** We explicitly included degraded scenes. For example, Figure 5 (P5_9) shows a direct comparison in a scene with heavy smoke. Even in this noisy condition, our method maintains better structural integrity and outperforms SOTA methods. (2) **More complex anatomy:** We also included scenes with complex anatomical structures. For instance, Figure 5 (P3_1) and Figure 17 (P3_5) show intricate tissue structures and tool-tissue interactions, where our method again demonstrates superior reconstruction fidelity. Therefore, we believe our comprehensive evaluation on all 35 sequences, which include diverse anatomies and degraded conditions, is a strong validation of our methodology's robustness.
>
> ---
> **Q3: Regarding computational overhead, runtime, and latency.**
>
> **A3:** We thank the reviewer for this critical point. We first clarify that the heavy models (SAM, MLLM) are used offline, one-time during data preprocessing to generate our target training data. They are not part of the training or inference loop. Our actual framework is efficient. We have added the following detailed benchmark table.
> Table A: Time and Computational Efficiency.
> | Training time | GPU Usage (infer) | Rendering speed | Retrieval speed | Retrieval Latency |
> | :--- | :--- | :--- | :--- | :--- |
> | 0.5h | 2.2GB | 130fps | 59.16fps | 16.90ms ||
>
> For computational overhead, our method follows the standard paradigm of 4DGS. However, our 30-minute training time is fast for this paradigm (which often requires several hours). This, combined with the low 2.2 GB GPU footprint, makes our system practical for near-real-time applications and highly accessible. For latency and temporal stability, the average retrieval latency is 16.90 ms, which corresponds to a 59.16 FPS frame rate for the full retrieval task. This performance is nearly 2x the 30 FPS real-time requirement of our source video and is well within the thresholds for smooth interactive applications. For long-term temporal stability: We provide a detailed analysis of this in Figure 16. As shown, our method maintains a stable, high-IoU retrieval over a very long sequence of 800+ frames. This is a strong test of temporal robustness, especially as many standard monocular dynamic scene datasets (e.g., HyperNeRF Vrig) are often shorter (e.g., ~500 frames).
>
> ---
> **Q4: Missing surgical SLAM citations.**
>
> **A4:** We thank the reviewer for pointing out this. We have cited these works in our related work section.

---

> ### Author Response · Authors · 2025-11-18
> **Response to Reviewer ADhH (3/3)**
>
> ---
> **Q5:  Regarding the generalizability of "open-vocabulary (OV)" retrieval.**
>
> **A5:** We thank the reviewer for raising this important point. We would like to clarify the precise scope of our OV claim. It refers to the ability to retrieve seen instruments using divers, natural-language phrasing, not just rigid templates. Our paper demonstrates this robustly in two key ways. First, our training process itself (Sec 3.2.1) is not based on rigid templates. We use an MLLM to enrich the basic semantics into diverse, natural-language sentences. We then train our model to match the embeddings from a powerful, pre-trained text encoder (BioBERT), which already possesses a rich understanding of synonyms and paraphrasing. Second, our evaluation protocol was specifically designed to validate this generalization, as we found that evaluation in prior work was often limited. Many existing methods for semantic scene understanding (e.g., 4DLangSplat, OpenGaussian, SurgTPGS) are often evaluated on simpler query types, such as single object names or basic object-state pairs. To truly test our model's robustness to unseen and complex phrasing, we propose a new, comprehensive evaluation protocol (detailed in Appendix D). This protocol systematically tests the model's robustness using four different query types. It includes those simpler queries (type-only, type+action) but also adds complex, multi-attribute natural language phrases. The state-of-the-art results in Table 3 are the average performance across all four of these diverse query types. This provides direct, quantitative proof that our model generalizes to the varied, natural-language phrasing the reviewer asked about, rather than just overfitting to templates.
>
> ---
> We thank the reviewer again for their constructive feedback. We believe the provision of comprehensive video demos, source code, and precise efficiency benchmarks fully resolves the concern regarding 'incomplete execution.' We hope these additions allow for a positive reassessment of our framework's strong potential.

---

### Official Review · Reviewer_2V1t · 2025-10-30

**Soundness:** 3
**Presentation:** 3
**Contribution:** 3
**Rating:** 6
**Confidence:** 3

**Summary:**

This paper introduces SurgRe4DGS, a new method for creating 4D models of surgical scenes from videos. The main idea is to combine 4D scene modeling using 3D Gaussian Splatting with language features. This allows the system to not only reconstruct the 3D scene as it changes over time, but also allows users to find specific surgical instruments using text-based search queries (e.g., "find the grasper on the right bottom"). A key contribution is the introduction of a new, large-scale dataset called StereoLung15K, which was created specifically for this combined task of 4D reconstruction and text-based retrieval. The authors show that their method performs better than existing methods on this new dataset and two other public datasets.

**Strengths:**

1. **Comprehensive Experimental Validation:** The paper's primary strength is its thorough experimental evaluation. The authors validate their method on three different datasets, StereoLung15K, StereoMIS, and EndoNerf. They compare against a large number of existing methods, including 11 baselines for reconstruction in Table 2, and 3 baselines for retrieval in Table 3, and demonstrate state-of-the-art results across all of them.
2. **Extensive Ablation Studies:** The authors provide a strong set of ablation studies in Sec 4.3 and Appendix E that clearly justify their design choices. They systematically test the impact of their spatial structural prior, the hybrid semantics injection, and even other parameters like top-K and the choice of language model. This makes the experimental results very convincing.
3. **Dataset Contribution:** The introduction of the StereoLung15K dataset is a major contribution to the field. As shown in Table 1, this appears to be the first large-scale dataset that provides annotations for both 4D reconstruction and text-driven tool retrieval in a surgical setting. This new resource will be valuable for future research.

**Weaknesses:**

1. **High Methodological Complexity:** The proposed method in Section 3 is very complex and combines many advanced techniques. It uses an MLLM to enrich text descriptions, a dual-branch mechanism, a hybrid semantic injection scheme, and contrastive learning. From the perspective of a non-expert (for me), it is difficult to follow the precise interactions between all these parts and pinpoint the most critical component responsible for the performance gain.
2. **Generalizability:** The new dataset focuses specifically on lung surgery (thoracoscopic procedures). While the method is also tested on other endoscopic datasets, it is unclear how well this approach would generalize to other, visually different types of surgery (e.g., open-field surgery) or to non-medical dynamic scenes.

**Questions:**

1. Could the authors provide a simplified, high-level diagram or explanation of how the visual features from SAM and the language features from the MLLM are combined in the "unified latent vector" and how this vector is used during retrieval?
2. The performance on the new StereoLung15K dataset is excellent. Do the authors believe this method is ready for clinical application, or what are the main practical barriers (e.g., computational speed, robustness) that still need to be addressed?

**Details Of Ethics Concerns:**

The paper includes a clear Ethical Statement and properly addresses data anonymization. They also include a statement on LLM usage at Appendix H, which is commendable.

---

> ### Author Response · Authors · 2025-11-18
> **Response to Reviewer 2V1t (1/2)**
>
> We sincerely thank Reviewer 2V1t for their thorough and highly supportive review. We are particularly encouraged that the reviewer recognized the "*comprehensive experimental validation*" of our work against extensive baselines, the "*strong set of ablation studies*" that justify our design choices, and the "*major contribution*" of our new StereoLung15K dataset to the field. We have addressed the reviewer's specific questions regarding methodological clarity and clinical application below.
>
> ---
> **Q1: Clarification on the critical components of the method.**
>
> **A1:** We thank the reviewer for this question regarding our method's components. As you noted, our ablation studies (in Sec 4.3 and Appendix E) are "*strong*" and "*convincing*," and they were designed precisely to provide this answer. The performance gain comes from a synergistic combination of critical contributions, each targeting a specific challenge in 4D surgical scenes:
>
> + For Reconstruction Fidelity: Our "Visual Representation Enhancement" mechanism is key. This is jointly validated by two ablations: (1) Introducing the "spatial structural prior" to learn visual features ($\mathcal{L}_v$) is crucial. Table 4 shows that adding this prior boosts PSNR by 1.014 on StereoMIS dataset. (2) Furthermore, using these learned visual features to enhance the Gaussian color representations (Eq. 6) further improves fidelity. Appendix E.1 (Figure 13) confirms this, showing this enhancement boosts PSNR on StereoMIS from 28.292 to 28.692.
> + For Retrieval Accuracy: The "Hybrid Semantics Injection Scheme" is the most critical component. Our ablation in Table 5 clearly quantifies this: our full hybrid model (with $\mathcal{L}\_{align}^{l}$ and $\mathcal{L}\_{cons}^{l}$) achieves 59.767 mIoU on StereoMIS, whereas models using only partial components (e.g., 44.635 mIoU for $\mathcal{L}_{align}^{l}$ only) perform significantly worse.
> + For 4D Temporal Consistency: The "temporal modulation" modules ($\Gamma_v$ and $\Gamma_l$) are essential for ensuring stability across time. As shown in Appendix E.2 (Figure 14), adding temporal modulation for the language features dramatically improves semantic stability, boosting mIoU on StereoMIS from 54.405 to 59.767.
> Therefore, our performance gain comes from this specific, experimentally-justified combination of design choices.
>
> ---
> **Q2: Generalizability of our method.**
>
> **A2:** We thank the reviewer for raising this important point. Our primary goal is to develop a generalizable framework for the broad domain of minimally invasive surgery (endoscopic, laparoscopic). As the reviewer noted, we tested on three datasets. We would like to elaborate on their significant diversity, as we believe this diversity provides strong evidence for the method's generalizability within our target domain: (1) EndoNerf: This dataset consists of DaVinci robotic prostatectomy (a pelvic procedure), which has different anatomy, motion patterns, and visual characteristics. (2) StereoMIS: This is a large-scale dataset for robotic-assisted abdominal/pelvic procedures (in porcine models) designed for SLAM, featuring complex tool-tissue interactions and breathing motion. (3) StereoLung15K (Ours): This is our new benchmark featuring binocular thoracoscopic (lung) surgery. It is the only dataset that supports the full joint task of 4D reconstruction and text-driven retrieval at 30 FPS. It also features extreme surgical complexities like bleeding, smoke, occlusions, and severe tissue deformations.
> We believe this provides strong validation for our methodology's generalizability within our target domain. Our framework consistently outperforms specialized SOTA methods across all three of these visually distinct and challenging surgical environments (thoracic, pelvic, and abdominal). To further address the reviewer's point on non-medical dynamic scenes, we conducted an additional experiment on a monocular video from the HyperNeRF dataset, comparing with the 4DGS baseline. As shown in the Table, our method achieved superior reconstruction results compared to the baseline 4DGS, demonstrating that our core dynamic modeling is robust. This provides strong additional evidence for our framework's methodological generalizability, showing its adaptability even to non-medical dynamic scenes.
>
> Table A: Quantitative Comparison Results on HyperNeRF Dataset.
> | | **Broom** | | **Chicken** | | **Peel Banana** | | **Mean** | |
> | :--- | :---: | :---: | :---: | :---: | :---: | :---: | :---: | :---: |
> | | **PSNR↑** | **SSIM↑** | **PSNR↑** | **SSIM↑** | **PSNR↑** | **SSIM↑** | **PSNR↑** | **SSIM↑** |
> | 4DGS | 21.53 | 0.351 | 26.82 | 0.797 | 27.82 | 0.844 |25.39 | 0.664 |
> | Ours | **21.892** | **0.398** | **26.851** | **0.805** | **27.904** | **0.859** | **25.549** | **0.687**|

---

> ### Author Response · Authors · 2025-11-18
> **Response to Reviewer 2V1t (2/2)**
>
> ---
> **Q3: Clarification on the vision-language latent vector and its usage in retrieval.**
>
> **A3:** We thank the reviewer for this question. The core idea is not to simply "mix" features in 3D space, but rather to use "parallel processing" and "decoupling" in the 2D rendering space. The simplified, high-level explanation is as follows.
> + **Step1:** Unified Representation. Instead of treating geometry and semantics separately, we attach a single unified feature vector to every 3D Gaussian point. This vector is simply a concatenation of two parts: a Visual part (for appearance) and a Language part (for semantics).
> + **Step 2:** Parallel Rendering. We use a parallel rasterizer to render these unified vectors onto the 2D screen simultaneously. This produces a single high-dimensional feature map that contains both visual and semantic information for the current view, ensuring perfect spatial alignment.
> + **Step 3:** Decoupling and Supervision. We then slice (decouple) this rendered map back into two distinct maps for supervision: (1) The Visual Map, which is trained to match structural features from SAM. (2) The Language Map, which is trained to match semantic features from the MLLM.
> + **Step 4:** Retrieval (Inference). For retrieval, we only need to render the Language Map. When a user provides a text query, we compute the similarity between the query's embedding and the rendered Language Map to generate the final segmentation mask.
>
> To clearly present our method, we have provided detailed description in Section 3.2.2 (following Eq. 3, "Furthermore, we...") and Appendix D (Open-Vocabulary 4D Instrument Retrieval) in the manuscript. Additionally, we have refined Figure 3 (pipeline) with a detailed caption to visually illustrate this framework structure.
>
> ---
> **Q4: On clinical application and practical barriers.**
>
> **A4:** We thank the reviewer for this insightful question about the practical path to clinical application. First, we would like to clarify that our method, like other SOTA 4D Gaussian Splatting methods, fundamentally operates as a scene-level optimization framework. This standard approach requires offline training to reconstruct each new surgical video with high fidelity. However, while this is a well-understood characteristic of the 4DGS field, our method was specifically designed to be exceptionally fast and efficient for clinical translation. As shown in the Table, our model requires only 0.5 hours (30 minutes) of training and uses only 2.2 GB of GPU VRAM during inference. This is a significant strength. Our 30-minute training time is fast for this paradigm (which often requires several hours). Furthermore, our inference performance is exceptional: the full retrieval task runs at 59.16 FPS (nearly 2x the 30 FPS real-time requirement) with a specific latency of only 16.90 ms. This is enabled by our efficient language branch, which adds minimal overhead to the fast ~130 FPS base rendering speed. Our real-time inference and low GPU footprint make it highly accessible. Therefore, we believe our method is immediately applicable and ready for all high-value offline clinical tasks, such as postoperative analysis, text-based search of surgical video archives, and developing advanced training simulators. While the 30-minute training time is not yet 'live', its speed makes our framework a highly promising path toward near-real-time clinical application. The primary remaining barrier is a common challenge for all methods in this field: maintaining robust reconstruction and retrieval performance in degraded scenes with extreme, unmodeled scenarios (like heavy smoke or occlusions).
>
> Table B: Time and Computational Efficiency.
> | Training time | GPU Usage (infer) | Rendering speed | Retrieval speed | Retrieval Latency |
> | :--- | :--- | :--- | :--- | :--- |
> | 0.5h | 2.2GB | 130fps | 59.16fps | 16.90ms |
>
> ---
> We thank the reviewer again for their strong support and insightful questions， and sincerely hope our responses reaffirm your positive assessment of our work.

---

### Official Review · Reviewer_W7vJ · 2025-11-02

**Soundness:** 2
**Presentation:** 2
**Contribution:** 3
**Rating:** 4
**Confidence:** 3

**Summary:**

The authors introduce SurgRe4DGS, a novel framework that jointly realizes 4D surgical scene reconstruction and open-vocabulary instrument retrieval through synergistic vision–language Gaussian representations. Particularly, they leverage visual clues to enhance color representations through spatial-color blending with temporal modulation, while language features are instilled with hybrid injection strategies via semantic map rendering and weighted top-K aggregation with contrastive learning. Extensive experiments on two public datasets, StereoMIS and EndoNerf, as well as the custom dataset StereoLung15K, demonstrate thatSurgRe4DGS achieves state-of-the-art performance in terms of reconstruction fidelity and retrieval precision.

**Strengths:**

1. The paper conduct extensive experiments, and the results are promising and show some improvement.
2. The supplementary section is helpful.
3. The research gap is addressed well.

**Weaknesses:**

1. I don't see the significant testing of this paper, and I won't be able to tell how significant these results are compare with current SOTA methods without the rigorous testing. I would recommend to perform such testings for improving the soundness of this paper.
2. Figure 3 is unnecessary.
3. Figure 4 is really cluttered and complicated with a lot of terms and parameters. I would expect the authors to do more through explanation for this paper instead of just summarizing it in couple of sentences.
5. The figure wrapped inside of the text is really hard to read and I believe it should be separated from the text.
6. Is there any failure cases (figures)? Why they are not being included in the paper for comparison and discussion?
7. Could the author make a separate table for the detailed meaning of the parameters used in the equation? It is hard to asset the accuracy/soundness of the provided equations without referring to a table for clarity since there are a lot of them needed to be tracked.

**Questions:**

The equations given are primarily adopted from prior arts, and can the authors specify what is the innovation part (either preprocessing or just a proof of the concept) in terms of the mathematical expression?

---

> ### Author Response · Authors · 2025-11-18
> **Response to Reviewer W7vJ (1/2)**
>
> Dear reviewer W7vJ, thank you for your time reviewing our manuscript and for providing constructive feedback. In the following, we respond to your questions in detail:
>
> **Q1: Significant testing of the experiments.**
>
> **A1:** Thank you for your valuable suggestion. To formally prove that our improvements are not due to random chance, we performed a paired t-test for both core tasks of our paper. We compare our method against the SOTA baselines using the per-scene mIoU scores (N=26) from our StereoLung15K dataset (detailed in Appendix Tables 9-12).
>
> Table A: Statistical Significance Testing for 4D Scene Reconstruction.
> | Baseline | PSNR | PSNR (Ours) | Improvement | $p$-value |
> |:---------:|:----:|:-----------:|:-----------:|:---------:|
> | EndoRDGS  |23.868|**24.524 dB**|  +0.656     |  1.6e-9   |
> | 4DGS      |23.231|**24.524 dB**|  +1.293     |  1.3e-11  |
>
> Table B: Statistical Significance Testing for Language-Driven Instrument Retrieval.
> | Baseline | mIoU | mIoU (Ours) | Improvement | $p$-value |
> |:---------:|:----:|:-----------:|:-----------:|:---------:|
> |4DLangSplat|43.914| **57.907**  |  +13.99     |  1.3e-7   |
> |   DGD     |49.884| **57.907**  |  +8.02      |  1.7e-4   |
>
> As shown in the table, all p-values are far below the 0.001 threshold. This confirms that our improvements in both 4D reconstruction and retrieval are statistically significant and not a result of random fluctuation. We will add the results of significant testing in the final version.
>
> **Q2: Clarification of Figure 4.**
>
> **A2:** Thanks for reviewer’s comment. We have uploaded a revised manuscript with the refined Figure 4 (now Figure 3) to better illustrate our method. Our framework is the first to jointly realize 4D surgical scene reconstruction and free-form language-driven instrument retrieval. The core of our method is a novel unified dual-branch Gaussian representation. We attach a latent vector $z_i = [f_i^v, f_i^l]$ to each Gaussian point and train these two branches synergistically: (1) The Dynamic Visual Branch: This branch distills 2D spatial-structural features (from SAM) and applies temporal modulation to ensure consistency, optimizing for high-fidelity 4D reconstruction via Color, Depth, and Visual feature losses. (2) The Language Branch: This branch first enriches semantics using an MLLM and then injects these features into the 3D Gaussians via our novel Hybrid Semantics Injection Scheme (combining rendering, Top-K alignment, and contrastive learning). (3) Inference Phase: Once trained, the model supports (a) 4D Reconstruction (rendering novel RGB-D views at any query time $t$) and (b) Open-Vocabulary Retrieval (computing cosine similarity between rendered language maps and text queries to localize targets).
>
> **Q3: Paper figures.**
>
> **A3:** Thanks for your feedback on presentation and readability. Our layout design aims to provide as many visualizations as possible to clearly illustrate the proposed method, all while working within the strict page limits. Based on the reviewer's suggestions, we have made corresponding adjustments in the revised manuscript to improve the overall readability and flow. Specifically, we have removed Figure 3 (word cloud) from the main body, and have optimized Figure 4 (pipeline) with a detailed caption to provide a clearer explanation of our framework.

---

> ### Author Response · Authors · 2025-11-18
> **Response to Reviewer W7vJ (2/2)**
>
> **Q4: Possible failure analysis and discussion.**
>
> **A4:** We thank the reviewer for this suggestion to improve the paper's soundness. We have presented some failure cases showing the limitations or reconstruction artifacts in the main paper and the Appendix. Our limitations are generally common challenges for the entire 4DGS field, particularly in complex surgical scenes. For example, reconstruction quality can be impacted by extremely fast, non-rigid motion, such as the breathing-induced tissue movement shown in Figure 17 (P5_2), which can cause some artifacts. Similarly, as discussed in the main paper, heavy smoke can degrade reconstruction, as seen in Figure 5 (P5_9). This is a difficult, common challenge, as the model attempts to reconstruct the semi-transparent smoke itself. However, we note that even in these degraded scenes, our method's visual-language priors provide robust guidance, allowing it to maintain better structural integrity and outperform SOTA methods. For retrieval, the performance is naturally downstream of the reconstruction quality, as our system is reconstruction-focused. When the underlying 4D reconstruction contains significant artifacts (like those from fast motion or smoke), this can lead to semantic drift in the language features, which impacts retrieval accuracy. An example of this can be seen in the artifacts of the Figure 19 retrieval mask.
> Therefore, a key future direction is to improve the upstream reconstruction robustness to these common degradations (fast motion, smoke), which will further enhance the stability of the semantic retrieval.
>
> **Q5: A detailed table for the meaning of the parameters.**
>
> **A5:** We thank the reviewer for the suggestion. We have provided this table in the Appendix. Please refer to the Appendix Table 13.
>
> **Q6: Clarification of mathematical innovation.**
>
> **A6:** We thank the reviewer for this question. Our core mathematical innovations lie in the novel mechanisms designed to handle the extreme dynamics and ambiguity of 4D surgical scenes: (1) Dual-Branch Temporal Modulation (Eq. 5 & 7): To ensure spatio-temporal consistency, we introduce modulation modules ($\Gamma_v$ and $\Gamma_l$) for both branches. These blend the base features with a temporal embedding ($PE(t)$), ensuring that the final rendered color and semantics are spatiotemporally smooth, which is critical for stable retrieval in dynamic surgery. (2) The Hybrid Semantics Injection Scheme: This is our primary contribution for retrieval accuracy. We designed a novel hybrid constraint that combines: a) Coarse Alignment: Standard semantic map rendering ($\mathcal{L}\_{render}^{l}$). b) Depth-Free Alignment ($\mathcal{L}\_{align}^{l}$): We introduce a weighted top-K aggregation loss. This provides a robust, depth-free method for semantic injection, bypassing the noisy surgical depth problem entirely. c) Disambiguation ($\mathcal{L}\_{cons}^{l}$): We introduce a cross-instance contrastive loss to explicitly solve the instrument confusion problem by forcing the features of different instruments to be distinct.
>
> ---
> Thank you once again for your review. We sincerely hope that our responses and the additional experimental results address all your concerns, and that these substantial improvements warrant a re-evaluation of our work.

---

### Note · Program_Chairs · 2026-01-21
**Submission Desk Rejected by Program Chairs**

This submission manipulated the official template to have larger margins.